# DiffGED: Computing Graph Edit Distance via Diffusion-based Graph Matching

## Abstract

Graph Edit Distance (GED), which aims to find an edit path with minimum number of edit operations to transform one graph into another, is a fundamental NP-hard problem and a widely used graph similarity measure. Recent matching-based hybrid approaches have demonstrated better scalability than A* search-based hybrids by reformulating GED as a graph matching problem. In these methods, a neural network predicts a single deterministic node matching matrix, from which top-$k$ node mappings are extracted iteratively to derive candidate edit paths. However, these methods often suffer from highly correlated candidates that easily lead to suboptimal solutions, while the iterative extraction becomes inefficient for large $k$. In this paper, we propose DiffGED, the first generative approach for GED computation. Specifically, we formulate the graph matching problem as a generative task, and employ a diffusion-based model to generate multiple diverse node matching matrices simultaneously, from which diverse node mappings can be efficiently extracted. The generative diversity introduced by the diffusion process enables DiffGED to avoid suboptimal solutions and achieve superior solution quality close to the exact solution. Experiments on real-world datasets show that DiffGED generates multiple diverse edit paths with accuracy comparable to exact solutions, while running faster than existing hybrid approaches. The source code is available at https://anonymous.4open.science/r/DiffGED-DF86.

## 1 Introduction

Graph Edit Distance (GED) is one of the most widely used similarity measures for graphs (Gouda & Arafa, 2015; Liang & Zhao, 2017; Bunke, 1997), with broad applications across computer vision and pattern recognition (Chen et al., 2020; Cho et al., 2013; Maergner et al., 2019). GED is defined as the minimum number of edit operations required to transform one graph into another. For instance, in Figure 1, transforming $G$ into $G'$ requires at least four edit operations, yielding $\text{GED}(G, G') = 4$. However, due to its NP-hard nature, traditional A* search methods (Neuhaus et al., 2006; Blumenthal & Gamper, 2020; Chang et al., 2020) struggle to scale even to graphs with only a few nodes, as the search space grows exponentially with the number of nodes (Blumenthal & Gamper, 2020). In contrast, matching-based methods (Riesen & Bunke, 2009; Bunke et al., 2011) formulate GED computation as a bipartite graph matching problem and can be solved in polynomial time, but they often yield solutions of unsatisfactory quality.

In recent years, there has been growing interest in combining deep learning with traditional methods to compute GED more effectively and efficiently. Current state-of-the-art methods (Piao et al., 2023; Cheng et al., 2025) adopt a class of hybrid approaches that aim to enhance the solution quality of matching-based methods. Specifically, given a pair of graphs, a neural network (e.g., GNNs) is trained to predict a single node matching matrix. From this matrix, the top-$k$ node mappings with maximum matching weights are then extracted iteratively as shown in Figure 2, where each extracted mapping corresponds to a candidate edit path, and the candidate edit path with the minimum number

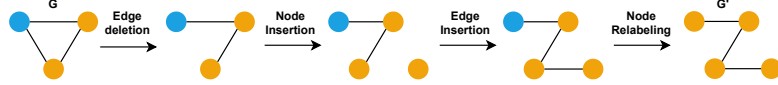

Figure 1: An optimal edit path for transforming $G$ to $G'$. $\text{GED}(G, G') = 4$.

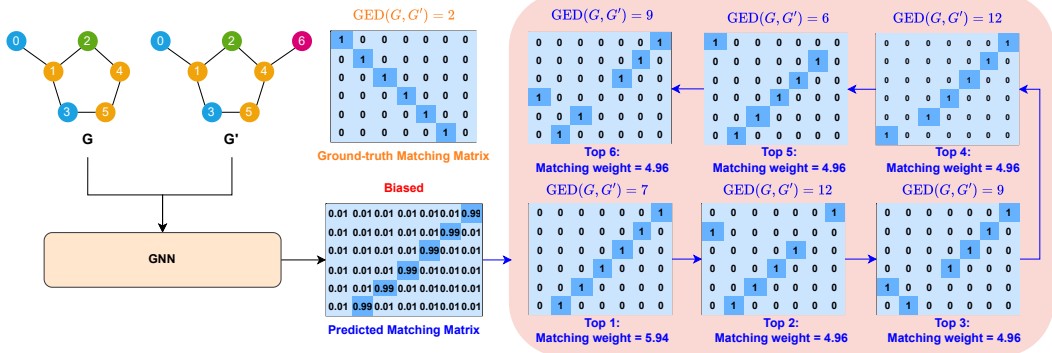

Figure 2: An example of existing matching-based hybrid approach that iteratively extracts top-$k$ maximum weight node mappings from a single deterministic node matching matrix predicted by GNN.

of edit operations is selected as the final solution. However, this approach is deterministic (i.e., for the same pair of graphs, it always produces the same deterministic output node matching matrix), and the extracted top-$k$ node mappings depend solely on a single predicted node matching matrix, with each mapping is extracted by searching on the previously extracted ones, leading to strong correlations among them. Thus, the following limitations could arise: (1) Highly correlated top-$k$ node mappings might easily fall into the local sub-optimal if the predicted node matching matrix is biased (i.e., significantly deviates from the correct matching). Consider the simple example of a biased predicted node matching matrix shown in Figure 2. It is clear to see that the top-6 node mappings extracted from the predicted matching matrix are highly correlated, and unfortunately, they are all sub-optimal with the derived GED significantly larger than the ground-truth GED; (2) Highly correlated node mappings limit the diversity of found edit paths, as multiple diverse edit paths could exist with multimodal distribution for an optimal GED; (3) The iterative extraction of top-$k$ node mappings is time consuming for large $k$, and cannot be parallelized to reduce the running time;

To address these limitations, we propose DiffGED, a novel generative approach that utilizes diffusion model for highly accurate GED computation. DiffGED first formulates matching-based GED computation as a generation task, then it generates $k$ diverse and high-quality node matching matrices in parallel from $k$ randomly initialized matrices, using our generative diffusion-based graph matching model DiffMatch. Next, $k$ candidate edit paths can be derived by extracting top-1 node mapping from each generated node matching matrix in parallel using a greedy algorithm. Therefore, comparing to previous deterministic approach, our proposed generative approach DiffGED offers the following advantages: (1) Each node mapping is extracted independently from a separate node matching matrix. With the stochasticity introduced by the generative diffusion model, the correlation between extracted node mappings is reduced, which enhances overall accuracy and decreases the likelihood of the extracted candidate solutions being trapped in local optima; (2) The reduced correlation further improves the diversity of the discovered edit paths; (3) Both the $k$ node matching matrices and their corresponding node mappings can be generated and extracted in parallel, which greatly reduces runtime when $k$ is large.

**Contributions.** To the best of our knowledge, we are the first to introduce a generative formulation for solving graph matching and GED computation. We are also the first to leverage a generative diffusion model for graph matching, namely DiffMatch. Extensive experiments on real-world datasets demonstrate that our proposed DiffGED (1) has exceptionally high accuracy (around $95\%$ on all datasets) which outperforms the existing methods by a great margin, (2) can generate diverse edit paths, and (3) has a shorter running time compared to other hybrid approaches.

## 2 RELATED WORK

**Traditional approaches.** Traditional approaches are often based on A* search (Neuhaus et al., 2006; Blumenthal & Gamper, 2020; Chang et al., 2020), guided by carefully designed heuristics to prune the unpromising search space. Unfortunately, these exact solvers are usually intractable

for large graphs due to the NP-hard nature of GED computation. To improve scalability, traditional matching-based approaches proposed to construct a node edition cost matrix, then model GED as a bipartite node matching problem and solve by either Hungarian (Riesen & Bunke, 2009) or VJ (Bunke et al., 2011) algorithm in polynomial time. However, the solution quality of matching-based methods are often poor.

**Regression-based deep learning approaches.** To address the limitations of traditional methods, deep learning approaches leverage the success of Graph Neural Networks (GNNs) in modeling complex graph structures. SimGNN (Bai et al., 2019) first formulated GED as a regression task with a cross-graph module, enabling fast and accurate prediction and inspiring many follow-ups (Bai & Zhao, 2021; Zhuo & Tan, 2022; Ling et al., 2021; Bai et al., 2020; Zhang et al., 2021; Qin et al., 2021; Jain et al., 2024; Li et al., 2019). However, these methods are not trained to recover edit paths, which are crucial in many applications (Wang et al., 2021), and their predictions may underestimate GED without corresponding feasible edit paths.

**Hybrid approaches.** To recover the edit paths, hybrid approaches have been extensively studied, combining traditional search-based methods with deep learning techniques. A well-studied line of research focuses on guiding A* search with heuristics learned by a neural network (Yang & Zou, 2021; Wang et al., 2021; Liu et al., 2023), aiming to improve the efficiency of the search process. However, these methods often suffer from poor solution quality and inherit the scalability limitations of A* search. To improve both efficiency and effectiveness, recent state-of-the-art hybrid approaches such as GEDGNN (Piao et al., 2023) and GEDIOT (Cheng et al., 2025) have shifted towards improving the solution quality of matching-based approaches. These methods work by predicting a single node matching matrix via neural network, from which top-$k$ node mappings can be iteratively extracted to construct candidate edit paths. Compared with A* search-based hybrid approaches, this class of methods is significantly more efficient. However, they are still ineffective, since all candidate edit paths are derived from the same deterministic matching matrix, they exhibit high correlation and are prone to local optima. Moreover, the iterative node mapping extraction process is inherently sequential and cannot be parallelized, leading to inefficiency for large $k$. Taken together, these challenges suggest substantial room for improvement in hybrid GED computation. To this end, we introduce a novel generative perspective that overcomes the inherent limitations of matching-based approaches and enables more effective and efficient GED computation.

## 3 PRELIMINARIES

In this paper, we focus on the computation of graph edit distance between a pair of undirected labeled graphs $G = (V, E, L)$ and $G' = (V', E', L')$, where $G$ consists of a set of nodes $V$, a set of edges $E$ and a labeling function $L$ that assigns each node a label.

**Graph Edit Distance (GED).**(Sanfeliu & Fu, 1983) Given a pair of graphs $(G, G')$, find an optimal edit path with minimum number of edit operations that transforms $G$ to $G'$. An edit path is a sequence of edit operations that transforms $G$ to $G'$. Graph edit distance $\text{GED}(G, G')$ is defined as the number of edit operations in the optimal edit path. Specifically, there are three types of edit operations: (1) insert or delete a node; (2) insert or delete an edge; (3) replace the label of a node.

**Edit path extraction.** Suppose $|V| \leq |V'|$, an edit path of transforming $G$ to $G'$ can be obtained from an injective node mapping $f$ from $V$ to $V'$ in linear time complexity $O(|V'| + |E| + |E'|)$ (Piao et al., 2023), such that $f(v) = v'$, where $v \in V$ and $v' \in V'$. The overall procedure can be described as follows:

(1) For each mapped node pair $f(v) = v'$, if $L(v) \neq L'(v')$, then replace the label of $v$ with $L'(v')$;

(2) For the remaining unmapped nodes in $V'$, insert $|V'| - |V|$ nodes into $V$. Each inserted node is mapped to and has the same label as an unmapped node in $V'$;

(3) For any two pairs of mapped nodes $f(v) = v'$ and $f(u) = u'$, if $(u, v) \in E$ and $(u', v') \notin E'$, delete the edge $(u, v)$ from $E$; if $(u, v) \notin E$ and $(u', v') \in E'$, insert the edge $(u, v)$ into $E$.

Therefore, to find an optimal edit path with minimum number of edit operations, we only have to find an optimal node mapping $f^*$. Due to the space limitation, the detailed algorithm can be found in Appendix B.1.

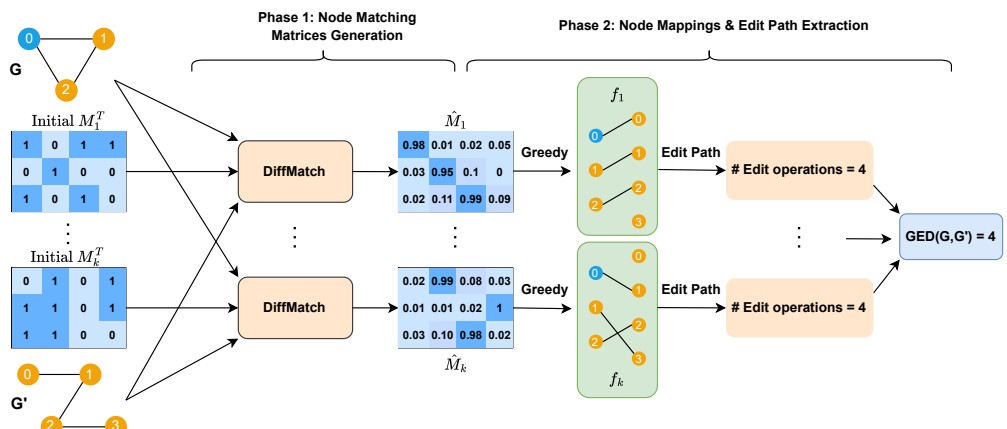

Figure 3: An overview of DiffGED. In the first phase, DiffGED first samples $k$ random initial node matching matrices, then DiffMatch will denoise the sampled node matching matrices via diffusion model. In the second phase, one node mapping will be extracted from each node matching matrix in parallel, and edit paths will be derived from the node mappings.

# 4  PROPOSED APPROACH: DIFFGED

## 4.1  DIFFGED: OVERVIEW

As described in Section 3, the optimal edit path can be obtained from an optimal node mapping $f^*$. To approximately find the optimal node mapping $f^*$, one simple and effective way is to predict top-$k$ node mappings $f_1, ..., f_k$, then select the one that results in the edit path with minimum edit operations.

To obtain top-$k$ node mappings, our DiffGED proposes a two-phase approach as shown in Figure 3. Specifically, in the first phase, given a graph pair $(G, G')$, instead of predicting a single node matching matrix, we predict top-$k$ node matching matrices $\hat{M}_1, ..., \hat{M}_k$ simultaneously, where each element in $\hat{M}_i \in \mathbb{R}^{|V| \times |V'|}$ represents the matching weight of a pair of nodes. Then, in the second phase, a simple greedy algorithm is used to extract top-1 node mapping independently from each predicted node matching matrix $\hat{M}_i$ in parallel, such that $f_i = Top1(\hat{M}_i)$. Comparing to existing matching-based approaches, our approach yields the following benefits: (1) Phase 1 reduces the correlation between each node mapping extracted in Phase 2, thus decreases the chances of falling into sub-optimal; (2) The reduced correlation naturally improves the diversity of the extracted node mappings; (3) Both the prediction of node matching matrices (Phase 1) and the extraction of node mappings (Phase 2) can be fully parallelized, significantly reducing the overall running time.

However, the neural networks in existing matching-based approaches cannot be easily adapted to Phase 1 of our approach. This is because they are deterministic and have limited capacity to predict a flexible number of node matching matrices for a given input graph pair. Once trained, they can only produce a fixed number of node matching matrices (often just one), and this requires a corresponding number of prediction heads in the network architecture, which consequently increases the number of unnecessary network parameters. Even worse, the node matching matrices predicted by different heads often remain highly correlated, as they share the same inputs and deterministic backbone, which inherently lack stochasticity.

To predict a flexible number of diverse node matching matrices, generative approach can be naturally well-suited to this improved two-phase approach. For Phase 1, we propose DiffMatch, a generative graph matching model that generates $k$ diverse and high-quality node matching matrices in parallel. As shown in Figure 3, unlike deterministic models that rely solely on the graph pair as input, our generative model DiffMatch introduces stochasticity by taking a randomly initialized discrete node matching matrix $M_i^T \in \{0, 1\}^{|V| \times |V'|}$ as an additional input. It then denoises the sampled $M_i^T$ to generate $\hat{M}_i \in \mathbb{R}^{|V| \times |V'|}$. This design enables the flexible generation of $k$ distinct node matching matrices in parallel by sampling $k$ random initial node matching matrices $M_1^T, ..., M_k^T$, with $k$ chosen arbitrarily at inference time and independent of the training phase. Therefore, it eliminates

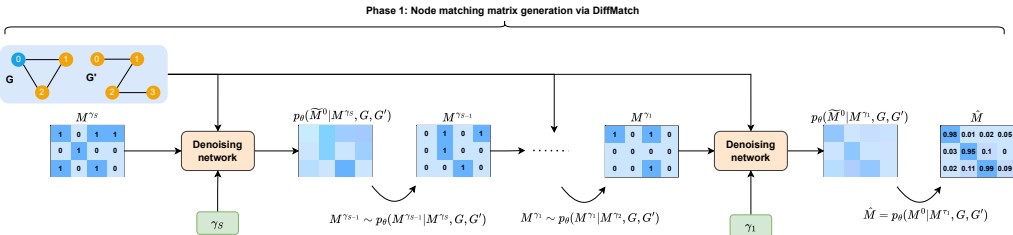

Figure 4: Reverse process of diffusion-based node matching model DiffMatch during inference.

the need for multiple prediction heads. Moreover, this generative formulation is also motivated by the fact that multiple optimal node mappings could exist with multimodal distribution for a given graph pair. Thus, different initial random node matching matrices can be mapped to different optima, consequently reducing the correlation among the generated node matching matrices.

To further enhance the generation of high-quality and diverse node matching matrices, our Diff-Match leverages the generative diffusion model (Ho et al., 2020; Dhariwal & Nichol, 2021; Sohl-Dickstein et al., 2015; Song & Ermon, 2019) to denoise each $M_i^T$, which has demonstrated impressive success in image generation tasks, but has not been explored in the context of graph matching. The main strength of diffusion model over other generative models is that it enables the generation of node matching matrices through an iterative refinement process, breaking down the complex generation task into simpler steps. Each step makes minor adjustments, progressively improving the quality of the matching matrices. Furthermore, each refinement step introduces stochasticity, which further reduces the correlation between generated node matching matrices and enhances the model's ability to produce diverse node matching matrices. To handle discrete data, we adopt discrete diffusion (Haefeli et al., 2022; Vignac et al., 2022; Austin et al., 2021) for DiffMatch.

## 4.2 PHASE 1: DIFFMATCH

In this section, we introduce our DiffMatch in detail based on a single discrete node matching matrix $M \in \{0,1\}^{|V| \times |V'|}$.

**Diffusion model overview.** Diffusion models are generative models that consist of a forward process and a reverse process. Given a ground-truth node matching matrix $M^0$ (transformed from the ground-truth node mapping), the forward process $q(M^{1:T}|M^0) = \prod_{t=1}^{T} q(M^t|M^{t-1})$ progressively corrupts $M^0$ to a sequence of increasingly noisy latent variables $M^{1:T} = M^1, M^2, ..., M^T$. Then, the reverse process learns to reconstruct $M^{t-1}$ from $M^t$ using a denoising network. During inference, the learned reverse process progressively denoises the latent variables towards the desired distribution, starting from a randomly sampled noise $M^T$, such that: $p_\theta(M^{0:T}|G, G') = p(M^T) \prod_{t=1}^{T} p_\theta(M^{t-1}|M^t, G, G')$.

**Forward process.** Let $\widetilde{M}^t \in \{0,1\}^{|V| \times |V'| \times 2}$ be the one-hot encoding of the node matching matrix $M^t$ at time step $t \in [0, T]$. The forward process adds noise to $M^{t-1}$ and samples $M^t$ from the following Categorical distribution: $q(M^t|M^{t-1}) = \text{Cat}(M^t|p = \widetilde{M}^{t-1}Q_t)$, with the transition probability matrix $Q_t = \begin{bmatrix} 1-\beta_t & \beta_t \\ \beta_t & 1-\beta_t \end{bmatrix}$, where $\beta_t$ is the probability of switching node matching state.

In practice, to sample the noisy matching matrix $M^t$ efficiently during training, we can compute the $t$-step marginal from $M^0$, such that: $q(M^t|M^0) = \text{Cat}(M^t|p = \widetilde{M}^0\overline{Q}_t)$, with $\overline{Q}_t = Q_1Q_2...Q_t$. Then, the denoising network is trained to predict node matching probabilities $p_\theta(\widetilde{M}^0|M^t, G, G')$ that reconstructs $M^0$ from $M^t$ by minimizing the binary cross-entropy loss (BCE):

$$\mathcal{L} = \frac{1}{|V||V'|} \sum_{v \in V} \sum_{v' \in V'} (M^0[v][v'] \log (p_\theta(\widetilde{M}^0|M^t, G, G')[v][v'])) + (1-M^0[v][v']) \log (1 - p_\theta(\widetilde{M}^0|M^t, G, G')[v][v']))$$

(1)

Due to the space limitation, the training procedure of the denoising network can be found in Appendix B.2.

Figure 5: An overview of the denoising network. The blue area denotes the network input, the yellow area denotes the architecture of the denoising network, and the pink area denotes the network output.

**Reverse process.** With the denoising network, each step $t$ of the reverse process can then denoise $M^t$ to $M^{t-1}$ as follows:

$$M^{t-1} \sim p_\theta(M^{t-1}|M^t, G, G') = \sum_{\widetilde{M}} q(M^{t-1}|M^t, \widetilde{M}^0) p_\theta(\widetilde{M}^0|M^t, G, G') \tag{2}$$

$$q(M^{t-1}|M^t, M^0) = \frac{q(M^t|M^{t-1}, M^0)q(M^{t-1}|M^0)}{q(M^t|M^0)} = \text{Cat}(M^{t-1}; p = \frac{\bar{M}^t Q_t^\top \odot \bar{M}^0 \overline{Q}_{t-1}}{\bar{M}^0 \overline{Q}_t (\bar{M}^t)^\top}) \tag{3}$$

where $q(M^{t-1}|M^t, M^0)$ denotes the posterior, with $\bar{M} \in \{0,1\}^{|V||V'| \times 2}$ obtained by reshaping $\widetilde{M} \in \{0,1\}^{|V| \times |V'| \times 2}$. During inference, starting from a random noisy discrete node matching matrix $M^T$, each $M_{t-1}$ can be sampled from $p_\theta(M^{t-1}|M^t, G, G')$ via Bernoulli sampling. And note that, for the last reverse step (i.e., $t-1=0$), we directly use $\hat{M} = p_\theta(M^{t-1}|M^t, G, G')$ as the input of the node mapping extraction in phase 2.

**Accelerating reverse process during Inference.** During training, the forward process typically employs a large number of steps $T$ (e.g., $T = 1000$), and performing $T$ reverse steps during inference can be computationally expensive. To accelerate DiffMatch's inference, we apply DDIM (Song et al., 2020) to the reverse process. The key idea of DDIM is that, instead of performing $T$ reverse steps over the entire sequence $[T, ..., 1]$, we perform only $S$ reverse steps on a sub-sequence $[\tau_S, ..., \tau_1]$ of $[T, ..., 1]$, where $S < T$ and $\tau_S = T$. We substitute $t$ and $t-1$ in Equation 2 with $\tau_i$ and $\tau_{i-1}$, and we rewritten the posterior as follows:

$$q(M^{\tau_{i-1}}|M^{\tau_i}, M^0) = \frac{q(M^{\tau_i}|M^{\tau_{i-1}}, M^0)q(M^{\tau_{i-1}}|M^0)}{q(M^{\tau_i}|M^0)} = \text{Cat}(M^{\tau_{i-1}}; p = \frac{\bar{M}^{\tau_i} \overline{Q}_{\tau_{i-1}, \tau_i}^\top \odot \bar{M}^0 \overline{Q}_{\tau_{i-1}}}{\bar{M}^0 \overline{Q}_{\tau_i} (\bar{M}^{\tau_i})^\top}) \tag{4}$$

where $\overline{Q}_{\tau_{i-1}, \tau_i} = Q_{\tau_{i-1}+1} Q_{\tau_{i-1}+2} ... Q_{\tau_i}$. The overall inference procedure of DiffMatch is presented in Figure 4, and a formal inference algorithm can be found in Appendix B.3.

**Denoising network.** An example of a 3-layer denoising network is shown in Figure 5. The network takes as input the graph pair $(G, G')$, the noisy node matching matrix $M^t$ along with its transpose $M^{t\top}$, and the corresponding time step $t$. Intuitively, it then works by directly computing the embeddings of each node matching pair, and predicting the node matching probabilities $p_\theta(\widetilde{M}^0|M^t, G, G')$ based on these embeddings to reconstruct $M^0$. Note that, $\text{GED}(G, G') = \text{GED}(G', G)$, we assume symmetry in node matching: if node $v \in V$ matches with node $v' \in V'$, then $v'$ also matches with $v$. Therefore, we only sample $M^t \in \mathbb{R}^{|V| \times |V'|}$ during both training and inference, then use both $M^t$ and $M^{t\top}$ as inputs to the denoising network.

For more details, let $\boldsymbol{h}_v^l$ and $\boldsymbol{h}_{v'}^l$ denote the embedding of node $v \in V$ and $v' \in V'$ at layer $l$, $\boldsymbol{h}_{vv'}^l$ and $\boldsymbol{h}_{v'v}^l$ denote the embedding of node matching pair $(v, v')$ and $(v', v)$ at layer $l$. For initialization, the node embeddings $\boldsymbol{h}_v^0$ and $\boldsymbol{h}_{v'}^0$ are initialized as the one-hot node labels, the node matching pair embeddings $\boldsymbol{h}_{vv'}^0$ and $\boldsymbol{h}_{v'v}^0$ are initialized as the sinusoidal embeddings (Vaswani et al., 2017) of

corresponding values in $M^t$ and $M^{t\top}$, and the time step embedding $\boldsymbol{h}_t$ is initialized as the sinusodial embedding of $t$.

For each layer $l$, the denoising network first updates the node embeddings of each graph to $\hat{\boldsymbol{h}}_v^l$ and $\hat{\boldsymbol{h}}_{v'}^l$, independently using their respective graph structures (intra-graph) via GIN (Xu et al., 2018). Then, the denoising network further refines the embeddings to $\boldsymbol{h}_v^l$ and $\boldsymbol{h}_{v'}^l$, while also updating the node matching pair embeddings to $\boldsymbol{h}_{vv'}^l$ and $\boldsymbol{h}_{v'v}^l$, by incorporating noisy interactions between node matching pairs (inter-graph) and the time step $t$ through Anisotropic Graph Neural Network (AGNN) (Joshi et al., 2020; Sun & Yang, 2023; Qiu et al., 2022). The key advantage of AGNN is its ability to directly updating embeddings for node matching pairs, enabling more expressive representations for cross-graph tasks. In contrast, traditional GNNs such as GIN are specifically designed for computing node embeddings only, making them less suited for capturing relationships between node pairs across graphs. Due to the space limitation, more details about AGNN can be found in Appendix B.4.

Finally, the denoising network computes the matching values of each node pair via multi-layer perceptron (MLP), and sums the matching values for corresponding pairs $(v, v')$ and $(v', v)$, then applies sigmoid activation to obtain the node matching probabilities $p_\theta(\widetilde{M}^0|M^t, G, G')$.

## 4.3 PHASE 2: NODE MAPPING EXTRACTION

After sampling $k$ noisy node matching matrices $M_1^T, ..., M_k^T$ and denoising to $\hat{M}_1, ..., \hat{M}_k$, we adopt the greedy algorithm based on matching weights to extract one node mapping from each node matching matrix. Specifically, assuming $|V| \leq |V'|$, the greedy node mapping extraction starts by selecting the node pair with the highest matching probability. Once a node pair is selected, all matching probabilities involving either of the selected nodes are set to $-\infty$ to prevent them from being selected again. This process is repeated iteratively $|V|$ times until every node in $V$ is assigned to a corresponding node in $V'$. Due to the space limitation, the detailed algorithm can be found in Appendix B.5.

Note that, the above greedy algorithm does not guarantee the extraction of optimal node mappings from the node matching matrices, but it has a time complexity of $O(|V|^2|V'|)$ slightly faster than the exact Hungarian algorithm (Kuhn, 1955) with time complexity of $O(|V'|^3)$. It can also be easily parallelized by GPU to extract $k$ node mappings from $k$ node matching matrices simultaneously to reduce the running time, especially for large $k$. It will be demonstrated in Appendix D.2 that DiffGED with the above greedy algorithm is sufficient to achieve excellent performance.

# 5 EXPERIMENTS

## 5.1 EXPERIMENTAL SETTINGS

**Datasets.** We conduct experiments over three popular real-world GED datasets: AIDS700 (Bai et al., 2019), Linux (Wang et al., 2012; Bai et al., 2019) and IMDB (Bai et al., 2019; Yanardag & Vishwanathan, 2015). For each dataset, we split $60\%$, $20\%$, and $20\%$ of all the graphs as training set, validation set, and testing set, respectively. To form training/validation/testing graph pairs, as well as their corresponding ground-truth labels, we follow the same strategy described in Piao et al. (2023). Due to space limitations, more details about datasets can be found in Appendix C.1.

**Baselines.** For traditional approximation methods, we compare our DiffGED with **Hungarian** (Riesen & Bunke, 2009) and **VJ** (Bunke et al., 2011). For A* search-based hybrid methods, we compare with: **Noah** (Yang & Zou, 2021), **GENN-A\*** (Wang et al., 2021), **MATA\*** (Liu et al., 2023). For matching-based hybrid methods, we compare with: **GEDGNN** (Piao et al., 2023), **GEDIOT**(Cheng et al., 2025). Due to the space limitation, details of each baseline can be found in Appendix C.2.

**Evaluation metrics.** We evaluate our DiffGED against other baseline methods based on the following metrics: (1) ***Mean Absolute Error (MAE)*** measures the average absolute difference between the predicted GED and the ground-truth GED; (2) ***Accuracy*** measures the ratio of the testing graph pairs with predicted GED equals to the ground-truth GED; (3) ***Spearman's Rank Correlation Coefficient ($\rho$)***, and (4) ***Kendall's Rank Correlation Coefficient ($\tau$)***, both measure the matching ratio between the ranking results of graphs based on their predicted GEDs and the ground-truth GEDs

Table 1: Overall performance on testing graph pairs. Methods with a running time exceeding 24 hours are marked with -.

| Datasets | Models | MAE ↓ | Accuracy ↑ | $\rho$ ↑ | $\tau$ ↑ | p@10 ↑ | p@20 ↑ | Time(s) ↓ |
|---|---|---|---|---|---|---|---|---|
| AIDS700 | Hungarian | 8.247 | 1.1% | 0.547 | 0.431 | 52.8% | 59.9% | 0.00011 |
| | VJ | 14.085 | 0.6% | 0.372 | 0.284 | 41.9% | 52% | 0.00017 |
| | Noah | 3.057 | 6.6% | 0.751 | 0.629 | 74.1% | 76.9% | 0.6158 |
| | GENN-A* | 0.632 | 61.5% | 0.903 | 0.815 | 85.6% | 88% | 2.98919 |
| | GEDGNN | 1.098 | 52.5% | 0.845 | 0.752 | 89.1% | 88.3% | 0.39448 |
| | GEDIOT | 1.188 | 53.5% | 0.825 | 0.73 | 88.6% | 86.7% | 0.39357 |
| | MATA* | 0.838 | 58.7% | 0.8 | 0.718 | 73.6% | 77.6% | **0.00487** |
| | DiffGED (ours) | **0.022** | **98%** | **0.996** | **0.992** | **99.8%** | **99.7%** | 0.0763 |
| Linux | Hungarian | 5.35 | 7.4% | 0.696 | 0.605 | 74.8% | 79.6% | 0.00009 |
| | VJ | 11.123 | 0.4% | 0.594 | 0.594 | 72.8% | 76% | 0.00013 |
| | Noah | 1.596 | 9% | 0.9 | 0.834 | 92.6% | 96% | 0.24457 |
| | GENN-A* | 0.213 | 89.4% | 0.954 | 0.905 | 99.1% | 98.1% | 0.68176 |
| | GEDGNN | 0.094 | 96.6% | 0.979 | 0.969 | 98.9% | 99.3% | 0.12863 |
| | GEDIOT | 0.117 | 95.3% | 0.978 | 0.966 | 98.8% | 99% | 0.13535 |
| | MATA* | 0.18 | 92.3% | 0.937 | 0.893 | 88.5% | 91.8% | **0.00464** |
| | DiffGED (ours) | **0.0** | **100%** | **1.0** | **1.0** | **100%** | **100%** | 0.06982 |
| IMDB | Hungarian | 21.673 | 45.1% | 0.778 | 0.716 | 83.8% | 81.9% | 0.0001 |
| | VJ | 44.078 | 26.5% | 0.4 | 0.359 | 60.1% | 62% | 0.00038 |
| | Noah | - | - | - | - | - | - | - |
| | GENN-A* | - | - | - | - | - | - | - |
| | GEDGNN | 2.469 | 85.5% | 0.898 | 0.879 | 92.4% | 92.1% | 0.42428 |
| | GEDIOT | 2.822 | 84.5% | 0.9 | 0.878 | 92.3% | 92.7% | 0.41959 |
| | MATA* | - | - | - | - | - | - | - |
| | DiffGED (ours) | **0.937** | **94.6%** | **0.982** | **0.973** | **97.5%** | **98.3%** | **0.15105** |

for each query testing graph; (5) ***Precision at top-*$10/20$ (p@$10/20$) measure the ratio of predicted top-$10/20$ similar graphs within the ground-truth top-$10/20$ similar graphs for each query testing graph; (6) ***Time(s)*** measures the average running time over all testing graph pairs.

**Implementation details.** Due to the space limitation, please refer to Appendix C.3.

## 5.2 MAIN RESULTS

Table 1 presents the overall performance of all methods on the test pairs. Across all datasets, DiffGED demonstrates exceptionally high solution quality in terms of MAE, accuracy, and all ranking metrics. For the AIDS700 dataset, the accuracy of DiffGED is nearly double that of other hybrid approaches. DiffGED consistently shows shorter running times than most hybrid approaches across all datasets, although it is slower than MATA* on smaller datasets. Note that, all A*-based hybrid approaches fail to complete evaluations (on IMDB) within a reasonable time due to the scalability issues inherent in A* search.

Specifically, both MATA* and DiffGED need to predict node matching matrices and then extract top-$k$ candidate results. However, they differ in key aspects: (1) MATA* predicts only two node matching matrices in a single step, whereas DiffGED generates $k$ node matching matrices in parallel over 10 denoising steps. This results in faster node matching matrix prediction for MATA*; (2) MATA* extracts the top-$k$ candidate matching nodes in $G'$ for each node in $G$, limiting the valid range of $k$ to $|V'|$ and typically selecting a small $k$ to reduce the A* search space. In contrast, DiffGED extracts the top-$k$ global maximum weight node mappings, allowing $k$ to be arbitrarily large. As a result, MATA* achieves shorter running times on smaller datasets. However, on larger datasets, MATA* suffers from the exponential growth of the A* search space, whereas DiffGED remains unaffected by this limitation.

Moreover, while GEDGNN and GEDIOT can scale to large graphs, they are both slower and perform worse across all datasets for several reasons. GEDGNN and GEDIOT iteratively extract top-$k$ candidate node mappings from a single predicted node matching matrix, resulting in highly correlated mappings. In contrast, DiffGED extracts top-$k$ candidate node mappings from $k$ different node matching matrices in parallel, generating diverse mappings. This diversity reduces the likelihood of falling into local sub-optimal solutions, even if some generated node matching matrices are biased. Additionally, the parallelization of node mapping extraction significantly reduces runtime.

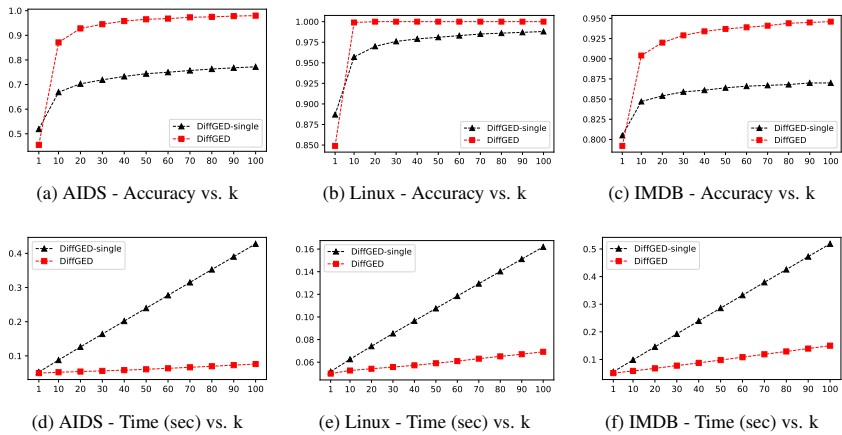

Figure 6: Effectiveness and Efficiency of Top-$k$ Approaches with Varying $k$.

## 5.3 ABLATION STUDY

**Generative top-$k$ approach.** To better evaluate the effectiveness, efficiency, and edit-path diversity of our generative top-$k$ approach, which extracts $k$ diverse node mappings from $k$ matching matrices, we compare it with the iterative approach commonly used in existing matching-based frameworks (e.g., GEDGNN, GEDIOT), which extracts highly correlated node mappings from a single node matching matrix. Specifically, we create a variant model, DiffGED-single, which generates only one node matching matrix using DiffMatch and then applies the iterative top-$k$ extraction.

As shown in Figure 6(a)-(c), our top-$k$ approach (DiffGED) performs slightly worse than DiffGED-single when $k = 1$. This is because DiffGED-single obtains the top-1 node mapping using the exact Hungarian algorithm, whereas DiffGED derives the top-1 mapping from the same node matching matrix via an approximate greedy algorithm. However, as $k$ increases, this initial disadvantage diminishes, with DiffGED rapidly converging to near-optimal accuracy, even with its approximate greedy algorithm. In contrast, DiffGED-single, despite using an exact extraction algorithm, converges to sub-optimal accuracy.

Notably, for simpler datasets like Linux, DiffGED achieves optimal solution quality with a small value of $k = 10$. The key reason behind this is that our generative approach generates a more diverse set of node mappings, which helps avoid sub-optimal solutions, whereas DiffGED-single's mappings tend to be highly correlated, leading to sub-optimal results. Moreover, even with iterative top-$k$ approach, it is interesting to note that DiffGED-single with $k = 100$ still achieves higher accuracy across all datasets compared to the results of GEDGNN and GEDIOT in Table 1, which highlights the effectiveness of our diffusion-based graph matching model Diff-Match. Furthermore, as shown in Figure 6(d)-(f), the running time of DiffGED-single increases significantly faster than that of DiffGED as $k$ grows. This disparity arises from DiffGED-single's iterative top-$k$ node mapping strategy, whereas DiffGED benefits from parallelized node matching matrix generation and parallel node mapping extraction. Since both processes in DiffGED are parallelized, the impact of increasing $k$ on its running time remains minimal, underscoring its superior efficiency for larger $k$ values.

Lastly, since multiple optimal edit paths often exist under a multimodal distribution, we evaluate edit paths diversity by computing the average number of distinct edit paths found per graph pair, where the number of edit operations is equal to the predicted minimum GED and the ground-truth GED, respectively, using $k = 100$. As demonstrated in Figure 8, our generative approach is capable of generating multiple distinct edit paths for both the predicted minimum GED and the ground-truth GED, while the iterative top-$k$ approach used in ex-

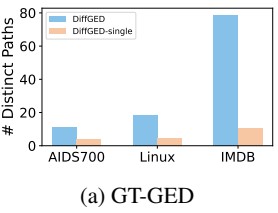

(a) GT-GED

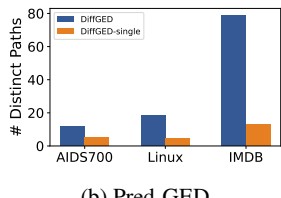

(b) Pred-GED

Figure 8: Evaluation of found edit path diversity. Pred-GED refers to average number of distinct edit paths with predicted minimum GED. GT-GED refers to average number of distinct edit paths with ground-truth GED.

isting matching-based approaches is limited to generating only a few. This further evident that our generative approach can generate diverse top-$k$ mappings, which enables us to effectively capture the multimodal distribution and avoid getting trapped in local optima. In contrast, the iterative approach used by existing frameworks produces highly correlated node mappings towards one mode, which limits its ability to capture the range of possible edit paths, thus could easily fall into sub-optimal results.

Due to the space limitation, more experimental results can be found in Appendix D.

## 6 CONCLUSION

This paper presents DiffGED, a novel generative framework for GED computation and edit path generation. Our generative approach works by generating $k$ diverse node-matching matrices simultaneously through our diffusion-based graph matching model, DiffMatch, and then extracting the top-$k$ node mappings in parallel using a greedy algorithm. Extensive experiments on real-world datasets demonstrate that our generative method outperforms all existing approaches by generating diverse, high-quality edit paths with accuracy close to $100\%$, all within a short running time.

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

## A  ADDITIONAL RELATED WORK

**Graph matching.**   Graph matching is a problem closely related to GED and deep-learning based graph matching has garnered significant attention across various domains, particularly in image feature matching (Jiang et al., 2022b; Wang et al., 2023; Chen et al., 2019; Jiang et al., 2022a). However, a fundamental distinction between the two problems lies in the nature of their ground truth. In graph matching, the ground truth is typically unique and application-specific, whereas in GED, multiple valid ground truths may exist due to different possible edit paths leading to the same graph transformation. Additionally, while graph matching focuses on maximizing node correspondence with respect to a predefined ground truth, GED aims to determine the minimal sequence of edit operations required to transform one graph into another. Another key difference lies in the characteristics of the input graphs. In graph matching, the input graphs are often structurally similar, whereas in GED, they can differ significantly. As a result, existing graph matching methods struggle to perform well in GED computation.

**Diffusion model.**   Diffusion models have emerged as a powerful class of generative models, achieving remarkable success in image generation and setting new benchmarks for high-quality image synthesis (Ho et al., 2020; Dhariwal & Nichol, 2021; Sohl-Dickstein et al., 2015; Song & Ermon, 2019). These models progressively refine random noise into structured outputs through a learned denoising process, demonstrating superior performance over traditional generative approaches such as GANs and VAEs. The success of diffusion models in continuous domains has inspired extensions to discrete data, leading to the development of discrete diffusion models for structured tasks, such as text generation (Austin et al., 2021). Building on these advancements, discrete diffusion has been extensively applied to graph generation (Vignac et al., 2022; Haefeli et al., 2022; Sun & Yang, 2023), where it has shown great potential in downstream tasks such as molecule generation and combinatorial optimization. This success further motivates the exploration of diffusion-based methods for a broader range of graph-related problems beyond generation.

## B  DETAILED METHOD

### B.1  EDIT PATH EXTRACTION

The detailed algorithm for edit path extraction with linear time complexity $O(|V'| + |E| + |E'|)$ is illustrated in Algorithm 1.

### B.2  TRAINING OF DIFFMATCH

The training procedure of the denoising network in our DiffMatch is outlined in Algorithm 2. For a given graph pair $(G, G')$ sampled from the training data with its ground-truth matching matrix $M^0$, we first sample a time step $t$ from a uniform distribution. Next, we sample a noisy matching matrix $M^t$ from the $t$-step marginal. Finally, the denoising network is trained to minimize the binary cross-entropy loss between the predicted matching matrix $p_\theta(\widetilde{M}^0 | M^t, G, G')$ and the ground-truth node matching matrix $\widetilde{M}^0$.

---

**Algorithm 1** Edit Path Generation

---

**Input:** $G = (V, E, L)$, $G' = (V', E', L')$, node mapping $f$;

1: $EditCost \leftarrow 0$;
2: **for** each $v \in V$ **do**
3:    **if** $L(v) \neq L'(f(v))$ **then**
4:       $L(v) \leftarrow L'(v')$;
5:       $EditCost \leftarrow EditCost + 1$;
6:    **end if**
7: **end for**
8: **for** each $v' \in V' \setminus \{f(v) \mid v \in V\}$ **do**
9:    Create a new $v$;
10:    $f(v) \leftarrow v'$ and $L(v) \leftarrow L'(v')$;
11:    $V \leftarrow V \cup \{v\}$;
12:    $EditCost \leftarrow EditCost + 1$;
13: **end for**
14: **for** each $(v, u) \in E$ **do**
15:    **if** $(f(v), f(u)) \in E'$ **then**
16:       $E \leftarrow E \setminus \{(v, u)\}$;
17:       $EditCost \leftarrow EditCost + 1$;
18:    **end if**
19: **end for**
20: **for** each $(v', u') \in E'$ **do**
21:    **if** $(f^{-1}(v), f^{-1}(u)) \notin E$ **then**
22:       $E \leftarrow E \cup \{(f^{-1}(v), f^{-1}(u))\}$;
23:       $EditCost \leftarrow EditCost + 1$;
24:    **end if**
25: **end for**
26: **return** $EditCost$;

---

**Algorithm 2** DiffMatch Training Procedure

---

**Input:** Graph pair $(G, G')$, Ground-truth node matching matrix $M^0$;

1: Sample $t \sim Uniform(1, ..., T)$;
2: Sample $M^t \sim q(M^t | M^0)$;
3: Take gradient step on $BCELoss(p_\theta(\widetilde{M}^0 | M^t, G, G'), M^0)$ via Equation 1;

---

### B.3 INFERENCE OF DIFFMATCH

Algorithm 3 illustrates the reverse process of DiffMatch during inference. During inference, starting from a noisy discrete node matching matrix $M^T$ randomly sampled from the Bernoulli distribution, each $M^{\tau_{i-1}}$ can be obtained from $p_\theta(M^{\tau_{i-1}} | M^{\tau_i}, G, G')$ via Bernoulli sampling. And for the last reverse step (i.e., $\tau_i = \tau_1$), we directly use $\hat{M} = p_\theta(M^0 | M^{\tau_1}, G, G')$ as the input of the node mapping extraction in phase 2.

---

**Algorithm 3** Sampling from DiffMatch

---

**Input:** Graph pair $(G, G')$, Random node matching matrix $M^T$;
1: **for** $\tau_i = \tau_S$ to $\tau_1$ **do**
2:    **if** $\tau_i \neq \tau_1$ **then**
3:       $M^{\tau_{i-1}} \sim p_\theta(M^{\tau_{i-1}} | M^{\tau_i}, G, G')$;
4:    **else**
5:       $\hat{M} \leftarrow p_\theta(M^0 | M^{\tau_1}, G, G')$;
6:    **end if**
7: **end for**
8: **return** $\hat{M}$;

---

**Algorithm 4** Greedy Node Mapping Extraction

---

**Input:** $i$-th node matching matrix $\hat{M}_i \in \mathbb{R}^{|V| \times |V'|}$;
**Output:** $i$-th node mapping $f_i$;
1: Initialize $f_i \leftarrow \emptyset$ ;
2: **for** $n \leftarrow 1$ to $|V|$ **do**
3:    select $(v, v')$ with the maximum value in $\hat{M}_i$;
4:    $f_i \leftarrow f_i \cup \{(v, v')\}$;
5:    set all elements in $v$-th row of $\hat{M}_i$ to $-\infty$;
6:    set all elements in $v'$-th column of $\hat{M}_i$ to $-\infty$;
7: **end for**
8: **return** $f_i$;

---

### B.4 ANISOTROPIC GRAPH NEURAL NETWORK

For each layer $l$ of our denoising network, the Anisotropic Graph Neural Network (AGNN) can be represented as follows:

$$
\begin{aligned}
\hat{\boldsymbol{h}}_{vv'}^l &= \boldsymbol{W}_1^l \boldsymbol{h}_{vv'}^{l-1}, \quad \hat{\boldsymbol{h}}_{v'v}^l = \boldsymbol{W}_1^l \boldsymbol{h}_{v'v}^{l-1} \\
\tilde{\boldsymbol{h}}_{vv'}^l &= \boldsymbol{W}_2^l \hat{\boldsymbol{h}}_{vv'}^l + \boldsymbol{W}_3^l \hat{\boldsymbol{h}}_v^l + \boldsymbol{W}_4^l \hat{\boldsymbol{h}}_{v'}^l \\
\tilde{\boldsymbol{h}}_{v'v}^l &= \boldsymbol{W}_2^l \hat{\boldsymbol{h}}_{v'v}^l + \boldsymbol{W}_3^l \hat{\boldsymbol{h}}_{v'}^l + \boldsymbol{W}_4^l \hat{\boldsymbol{h}}_v^l \\
\boldsymbol{h}_{vv'}^l &= \hat{\boldsymbol{h}}_{vv'}^l + \text{MLP}^l(\text{ReLU}(\text{GN}_{MM^\top}(\tilde{\boldsymbol{h}}_{vv'}^l)) + \boldsymbol{W}_5^l \boldsymbol{h}_t) \\
\boldsymbol{h}_{v'v}^l &= \hat{\boldsymbol{h}}_{v'v}^l + \text{MLP}^l(\text{ReLU}(\text{GN}_{MM^\top}(\tilde{\boldsymbol{h}}_{v'v}^l)) + \boldsymbol{W}_5^l \boldsymbol{h}_t) \\
\boldsymbol{h}_v^l &= \hat{\boldsymbol{h}}_v^l + \text{ReLU}(\text{GN}_{GG'}(\boldsymbol{W}_6^l \hat{\boldsymbol{h}}_v^l + \sum_{v' \in V'} \boldsymbol{W}_7^l \hat{\boldsymbol{h}}_{v'}^l \odot \sigma(\tilde{\boldsymbol{h}}_{vv'}^l))) \\
\boldsymbol{h}_{v'}^l &= \hat{\boldsymbol{h}}_{v'}^l + \text{ReLU}(\text{GN}_{GG'}(\boldsymbol{W}_6^l \hat{\boldsymbol{h}}_{v'}^l + \sum_{v \in V} \boldsymbol{W}_7^l \hat{\boldsymbol{h}}_v^l \odot \sigma(\tilde{\boldsymbol{h}}_{v'v}^l)))
\end{aligned}
\tag{5}
$$

where $\boldsymbol{W}_1^l, \boldsymbol{W}_2^l, \boldsymbol{W}_3^l, \boldsymbol{W}_4^l, \boldsymbol{W}_5^l, \boldsymbol{W}_6^l, \boldsymbol{W}_7^l$ are learnable parameters at layer $l$, $\text{MLP}^l$ denotes multi-layer perceptron at layer $l$, $\text{GN}_{MM^\top}$ is the graph normalization (Cai et al., 2021) over all node matching pairs in both $M^t$ and $M^{t\top}$, $\text{GN}_{GG'}$ is the graph normalization over all nodes in both $G$ and $G'$, and $\sigma$ is the sigmoid activation.

### B.5 PHASE 2: NODE MAPPING EXTRACTION

Given a predicted node matching matrix $\hat{M}_i$, Algorithm 4 outlines the overall greedy procedure to extract top-1 node mapping from $\hat{M}_i$.

## C DETAILED EXPERIMENTAL SETTINGS

### C.1 DATASETS

We conduct experiments over three popular real-world GED datasets: AIDS700 (Bai et al., 2019), Linux (Wang et al., 2012; Bai et al., 2019) and IMDB (Bai et al., 2019; Yanardag & Vishwanathan, 2015). Each graph in AIDS700 is labeled, while each graph in Linux and IMDB is unlabeled.

Table 2: Dataset description.

| Dataset | # Graphs | Avg $|V|$ | Avg $|E|$ | Max $|V|$ | Max $|E|$ | Number of Labels |
|---------|----------|-----------|-----------|-----------|-----------|------------------|
| AIDS700 | 700 | 8.9 | 8.8 | 10 | 14 | 29 |
| Linux | 1000 | 7.6 | 6.9 | 10 | 13 | 1 |
| IMDB | 1500 | 13 | 65.9 | 89 | 1467 | 1 |

The statistics of datasets are summarized in Table 2. We obtain the ground-truth edit path (node mappings) from Piao et al. (2023). However, the ground-truth GED and edit paths are often computationally expensive to obtain for graph pairs with at least one graph has more than 10 nodes. To handle this, we follow the same strategy as described in Piao et al. (2023) to generate synthetic graphs for IMDB dataset. Specifically, for each graph $G$ with more than 10 nodes, synthetic graphs are generated by randomly applying $\Delta$ edit operations to $G$, these random edit operations are used as an approximation of the ground-truth edit path and $\Delta$ is used as an approximate of ground-truth GED. For graphs with more than 20 nodes, $\Delta$ is randomly distributed in $[1, 10]$, for graphs with more than 10 nodes and less than 20 nodes, $\Delta$ is randomly distributed in $[1, 5]$.

For each dataset, we split 60%, 20%, and 20% of all the graphs as training set, validation set, and testing set, respectively. To form training pairs, each training graph with no more than 10 nodes is paired with all other training graphs with no more than 10 nodes, each training graph with more than 10 nodes is paired with 100 synthetic graphs. In the validation and testing sets, each graph with no more than 10 nodes is paired with 100 random training graphs with no more than 10 nodes, and each graph with more than 10 nodes is paired with 100 synthetic graphs.

### C.2 DETAILS OF BASELINE METHODS

We compare our DiffGED with the following hybrid frameworks: (1) **Noah** (Yang & Zou, 2021) proposed using a pre-trained Graph Path Network (GPN) as the heuristic for A* beam search; (2) **GENN-A*** (Wang et al., 2021) introduced a Graph Edit Neural Network (GENN) to guide A* search by dynamically predicting the edit costs of unmatched subgraphs; (3) **MATA*** (Liu et al., 2023) proposed to prune the search space of A* search by extracting top-$k$ candidate matches for each node from two predicted node matching matrices; (4) **GEDGNN** (Piao et al., 2023) predicts a single deterministic node matching matrix, then iteratively extracts top-$k$ node mappings and edit paths; (5) **GEDIOT**(Cheng et al., 2025) follows the same approach as GEDGNN and further improves the prediction of node matching matrix via optimal transport.

### C.3 IMPLEMENTATION DETAILS

During training of our DiffMatch, we set the number of time steps $T$ to $1,000$ with linear noise schedule, where $\beta_0 = 10^{-4}$ and $\beta_T = 0.02$. For the reverse denoising process during testing, we set the number of time steps $S$ to 10 with linear denoising schedule, and we generate $k = 100$ node matching matrices in parallel for each testing graph pair.

For our denoising network, we set the number of layers to 6, the output dimension of each layer is 128, 64, 32, 32, 32, 32, respectively. We train it for 200 epochs with batch size of 128, we adopt Adam optimizer (Kingma, 2014) with learning rate of 0.001 and weight decay of $5 \times 10^{-4}$.

All experiments are conducted using Nvidia Geforce RTX3090 24GB and Intel i9-12900K with 128GB RAM.

## D MORE EXPERIMENTAL RESULTS

### D.1 GENERALIZATION ABILITY

**Generalization on unseen graph pairs.** To evaluate the generalization ability to unseen graphs of our DiffGED, instead of pairing each testing graph with 100 graphs from the training set, we pair each testing graph with 100 unseen graphs from the testing set. Table 3 presents the overall performance of all methods on these unseen testing graph pairs. Compared to the results in Table 1,

Table 3: Overall performance on unseen testing graph pairs. Methods with a running time exceeding 24 hours are marked with -.

| Datasets | Models | MAE ↓ | Accuracy ↑ | $\rho$ ↑ | $\tau$ ↑ | p@10 ↑ | p@20 ↑ | Time(s) ↓ |
|---|---|---|---|---|---|---|---|---|
| AIDS700 | Hungarian | 8.237 | 1.5% | 0.527 | 0.416 | 54.3% | 60.3% | 0.0001 |
| | VJ | 14.171 | 0.9% | 0.391 | 0.302 | 44.9% | 52.9% | 0.00016 |
| | Noah | 3.174 | 6.8% | 0.735 | 0.617 | 77.8% | 76.4% | 0.5765 |
| | GENN-A* | 0.508 | 67.1% | 0.917 | 0.836 | 87.1% | 90.6% | 3.44326 |
| | GEDGNN | 1.155 | 50.5% | 0.838 | 0.746 | 89.1% | 87.6% | 0.39344 |
| | GEDIOT | 1.348 | 47.4% | 0.81 | 0.71 | 88.4% | 86.9% | 0.39707 |
| | MATA* | 0.885 | 56.6% | 0.77 | 0.689 | 73.2% | 76.6% | **0.00486** |
| | DiffGED (ours) | **0.024** | **96.4%** | **0.993** | **0.986** | **99.7%** | **99.7%** | 0.07546 |
| Linux | Hungarian | 5.423 | 7.5% | 0.725 | 0.623 | 75% | 77% | 0.00008 |
| | VJ | 11.174 | 0.4% | 0.613 | 0.512 | 70.6% | 74.5% | 0.00013 |
| | Noah | 1.879 | 8% | 0.872 | 0.796 | 84.3% | 92.2% | 0.25712 |
| | GENN-A* | 0.142 | 92.9% | 0.976 | 0.94 | 99.6% | 99.6% | 1.17702 |
| | GEDGNN | 0.105 | 96.2% | 0.979 | 0.968 | 98.6% | 98.5% | 0.12169 |
| | GEDIOT | 0.14 | 94.8% | 0.973 | 0.959 | 98.1% | 98.3% | 0.12826 |
| | MATA* | 0.201 | 91.5% | 0.948 | 0.903 | 86.2% | 90.2% | **0.00464** |
| | DiffGED (ours) | **0.0** | **100%** | **1.0** | **1.0** | **100%** | **100%** | 0.06901 |
| IMDB | Hungarian | 21.156 | 45.9% | 0.776 | 0.717 | 84.2% | 82.1% | 0.00012 |
| | VJ | 44.072 | 26.6% | 0.4 | 0.359 | 60.1% | 63.1% | 0.00037 |
| | Noah | - | - | - | - | - | - | - |
| | GENN-A* | - | - | - | - | - | - | - |
| | GEDGNN | 2.484 | 85.5% | 0.895 | 0.876 | 92.3% | 91.7% | 0.42662 |
| | GEDIOT | 2.83 | 84.4% | 0.989 | 0.876 | 92.5% | 92.4% | 0.42269 |
| | MATA* | - | - | - | - | - | - | - |
| | DiffGED (ours) | **0.932** | **94.6%** | **0.982** | **0.974** | **97.5%** | **98.4%** | **0.15107** |

Table 4: Overall Performance on IMDB testing graph pairs. IMDB-small refers to training set that only contains real small graph pairs. IMDB-mix refers to training set that contains a combination of real small graph pairs and synthetic large graph pairs.

| Training set | Models | MAE ↓ | Accuracy ↑ | $\rho$ ↑ | $\tau$ ↑ | p@10 ↑ | p@20 ↑ | Time(s) ↓ |
|---|---|---|---|---|---|---|---|---|
| IMDB-small | GEDGNN | 7.943 | 77.1% | 0.844 | 0.815 | 88.2% | 87.6% | 0.48253 |
| | GEDIOT | 7.761 | 76.8% | 0.86 | 0.827 | 90.5% | 89.9% | 0.473 |
| | DiffGED | 5.789 | 83% | 0.892 | 0.874 | 90.1% | 90.8% | 0.14923 |
| IMDB-mix | GEDGNN | 2.469 | 85.5% | 0.898 | 0.879 | 92.4% | 92.1% | 0.42428 |
| | GEDIOT | 2.822 | 84.5% | 0.9 | 0.878 | 92.3% | 92.7% | 0.41959 |
| | DiffGED | 0.937 | 94.6% | 0.982 | 0.973 | 97.5% | 98.3% | 0.15105 |

it demonstrates that DiffGED can still achieve superior performance without losing accuracy, even with more challenging unseen testing graph pairs.

**Generalization on large graphs.** Moreover, in real-world scenarios, obtaining ground-truth node mappings for large graph pairs is often impractical. To evaluate the generalization ability of DiffGED under such conditions, we modify the training setup. Instead of training each method on a combination of real small graph pairs and synthetic large graph pairs from IMDB, we train each method exclusively on real small graph pairs from IMDB. However, the testing set still consists of a combination of real small graph pairs and synthetic large graph pairs. Table 4 presents the overall performance of DiffGED, GEDGNN and GEDIOT when trained on real small graph pairs. As observed, the accuracy of both DiffGED, GEDGNN and GEDIOT degrades, primarily because the testing graph pairs differ from the training graph pairs not only in graph size but also in distribution, due to the presence of synthetic graph pairs in the testing set, as these synthetic graphs differ from real graph pairs. Despite this challenge, DiffGED still outperforms GEDGNN and GEDIOT, achieving higher accuracy.

**Generalization on datasets without structural train–test leakage.** In addition, the AIDS, Linux, and IMDB datasets have recently been shown to suffer from structural train–test leakage (Roy et al.), meaning that a significant proportion of graphs in these datasets are isomorphic. This leakage may cause the reported results to overestimate the true generalization ability of each method. To address this concern, we follow the procedure described in (Roy et al.) to remove all isomorphic graphs to obtain unique graphs, and then form training and testing pairs using only these unique graphs. Table 5 shows the results of each method on datasets without structural train-test leakage. It is clear to see that after removing train-test leakage, our DiffGED can still achieve near-optimal performance

Table 5: Overall performance without structural train-test leakage.

| Datasets | Setting | Models | MAE ↓ | Accuracy ↑ | ρ ↑ | τ ↑ | p@10 ↑ | p@20 ↑ | Time(s) ↓ |
|---|---|---|---|---|---|---|---|---|---|
| AIDS700 | Cross-train-test | GEDGNN | 1.148 | 51.8% | 0.836 | 0.742 | 88.9% | 88.2% | 0.39227 |
| | | GEDIOT | 1.159 | 53.8% | 0.832 | 0.737 | 89.6% | 89% | 0.39381 |
| | | DiffGED (ours) | **0.046** | **96%** | **0.992** | **0.983** | **99.8%** | **99.6%** | **0.07431** |
| | Intra-test | GEDGNN | 1.235 | 48.7% | 0.824 | 0.729 | 90.1% | 88.4% | 0.39079 |
| | | GEDIOT | 1.349 | 46.5% | 0.8 | 0.701 | 88.1% | 86.9% | 0.39263 |
| | | DiffGED (ours) | **0.064** | **94.4%** | **0.987** | **0.975** | **99.5%** | **99.5%** | **0.07438** |
| Linux | Cross-train-test | GEDGNN | 0.935 | 65.1% | 0.809 | 0.722 | 85.8% | 87.4% | 0.27418 |
| | | GEDIOT | 1.009 | 65.3% | 0.788 | 0.706 | 87.9% | 85% | 0.27556 |
| | | DiffGED (ours) | **0.165** | **92.4%** | **0.958** | **0.931** | **93.7%** | **95.3%** | **0.07277** |
| | Intra-test | GEDGNN | 1.335 | 57.6% | 0.755 | 0.664 | 85.3% | 100% | 0.28935 |
| | | GEDIOT | 1.435 | 52.6% | 0.772 | 0.686 | 86.3% | 100% | 0.29775 |
| | | DiffGED (ours) | **0.305** | **86.1%** | **0.896** | **0.857** | **92.1%** | **100%** | **0.07694** |
| IMDB | Cross-train-test | GEDGNN | 4.799 | 73.1% | 0.817 | 0.783 | 85.2% | 85.5% | 0.75584 |
| | | GEDIOT | 4.679 | 74.9% | 0.826 | 0.794 | 87.6% | 86.8% | 0.73493 |
| | | DiffGED (ours) | **1.12** | **94%** | **0.973** | **0.963** | **97.1%** | **97.1%** | **0.22247** |
| | Intra-test | GEDGNN | 4.822 | 73.1% | 0.822 | 0.789 | 85.9% | 86.1% | 0.75577 |
| | | GEDIOT | 4.689 | 74.8% | 0.829 | 0.797 | 87.9% | 87% | 0.74122 |
| | | DiffGED (ours) | **1.141** | **93.8%** | **0.971** | **0.961** | **97%** | **97.2%** | **0.22315** |

on all datasets, whereas the performance of other baseline methods downgrades significantly. This again demonstrates the strong generalization ability of our DiffGED.

## D.2 ABLATION STUDIES

Table 6: Ablation study on testing graph pairs.

| Datasets | Models | MAE ↓ | Accuracy ↑ | ρ ↑ | τ ↑ | p@10 ↑ | p@20 ↑ | Time(s) ↓ |
|---|---|---|---|---|---|---|---|---|
| AIDS700 | DiffGED | 0.022 | 98% | 0.996 | 0.992 | 99.8% | 99.7% | 0.0763 |
| | DiffGED(w/o diffusion) | 1.618 | 46.7% | 0.732 | 0.629 | 82.4% | 81.1% | 0.01179 |
| | GEDGNN | 1.098 | 52.5% | 0.845 | 0.752 | 89.1% | 88.3% | 0.39448 |
| | GEDGNN(AGNN) | 0.736 | 66.7% | 0.884 | 0.812 | 94% | 93.1% | 0.39112 |
| Linux | DiffGED | 0.0 | 100% | 1.0 | 1.0 | 100% | 100% | 0.06982 |
| | DiffGED(w/o diffusion) | 0.743 | 74.7% | 0.887 | 0.839 | 96.4% | 95.8% | 0.01117 |
| | GEDGNN | 0.094 | 96.6% | 0.979 | 0.969 | 98.9% | 99.3% | 0.12863 |
| | GEDGNN(AGNN) | 0.061 | 97.4% | 0.992 | 0.987 | 99.6% | 99.5% | 0.13164 |
| IMDB | DiffGED | 0.937 | 94.6% | 0.982 | 0.973 | 97.5% | 98.3% | 0.15105 |
| | DiffGED(w/o diffusion) | 0.832 | 93.3% | 0.942 | 0.93 | 98.6% | 96.8% | 0.01944 |
| | GEDGNN | 2.469 | 85.5% | 0.898 | 0.879 | 92.4% | 92.1% | 0.42428 |
| | GEDGNN(AGNN) | 1.766 | 89.1% | 0.903 | 0.89 | 93.9% | 92.8% | 0.41387 |

**Do we really need diffusion?** The core idea of the proposed framework is to generate diverse, high-quality node matching matrices through an iterative reverse process of the diffusion model. To assess the effectiveness of the diffusion model in DiffMatch, we introduce a one-shot generative variant model, DiffGED(w/o diffusion), which takes a graph pair and a randomly initialized node matching matrix as input and directly predicts the clean node matching matrix, followed by greedy node mapping extraction. In this setup, we remove the time step component from the denoising network. During training, DiffGED(w/o diffusion) is also provided with a random node matching matrix instead of a noisy node matching matrix sampled from the forward diffusion process.

Table 6 presents the overall performance of DiffGED(w/o diffusion). Notably, DiffGED(w/o diffusion) performs poorly, and its performance is even worse than GEDGNN and GEDIOT on the AIDS and Linux datasets.

From a solution quality perspective, DiffGED(w/o diffusion) attempts to generate a high-quality node matching matrix in a single step from random noise, making the learning task extremely challenging. In contrast, the diffusion model decomposes this complex generation task into simpler, iterative refinements. The reverse diffusion process gradually denoises the random node matching matrix step by step, ensuring that each step only requires minor corrections. This progressive refinement leads to higher-quality node matching matrices.

From a solution diversity perspective, DiffGED introduces stochasticity at each reverse step during inference, whereas the stochasticity in DiffGED(w/o diffusion) comes solely from the random noise input. As a result, DiffGED is more likely to generate diverse node matching matrices. Furthermore, in diffusion models, the training input consists of a ground-truth node matching matrix corrupted by the forward diffusion process, rather than pure noise, and noisy matching matrix is only mapped to the ground-truth matching matrix. However, in DiffGED(w/o diffusion), the training input is pure

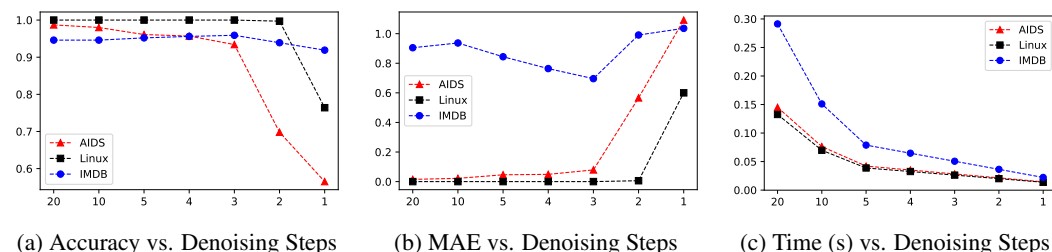

(a) Accuracy vs. Denoising Steps    (b) MAE vs. Denoising Steps    (c) Time (s) vs. Denoising Steps

Figure 9: Performance comparison across different reverse denoising steps during inference

noise, requiring a single random noise to map to multiple ground-truth matching matrices. This one-to-many mapping increases the likelihood of mode collapse, reducing the model's ability to generate diverse solutions. Therefore, diffusion model is necessary for our DiffGED to generate high quality and diverse node matching matrices. But it is interesting to note that the running time of DiffGED(w/o diffusion) is much shorter than DiffGED since it generates node matching matrices in one-shot without iteration.

**Anisotropic Graph Neural Network.** Instead of computing only node embeddings and then using their inner product to predict node matching probabilities, our denoising network leverages the Anisotropic Graph Neural Network (AGNN) to directly compute node pair embeddings, enabling a more expressive prediction of node matching probabilities.

To evaluate the effectiveness of AGNN, we create a variant of GEDGNN, GEDGNN(AGNN), that replaces its Cross Matrix Module with AGNN (without time steps). Moreover, we initialize a fixed node matching matrix filled with ones as input of GEDGNN(AGNN). We choose to create a variant of GEDGNN rather than creating a variant of DiffMatch by replacing AGNN with the Cross Matrix Module. This is because DiffMatch requires a noisy node matching matrix as input, but the Cross Matrix Module of GEDGNN ($\text{MLP}([h_v^\top W_1 h_{v'}, ..., h_v^\top W_c h_{v'}])$) cannot incorporate such noisy information when computing node matching probabilities. This limitation makes Cross Matrix Module unsuitable for direct integration into DiffMatch, leading us to use GEDGNN(AGNN) as the evaluation model for AGNN instead.

The overall performance of GEDGNN(AGNN) is presented in Table 6. The performance of GEDGNN increased significantly by incorporating AGNN, demonstrating that AGNN effectively enhances the model's ability to predict node matching probabilities by directly computing expressive node pair embeddings.

**Varying Reverse Denoising Steps during Inference.** During inference, DiffMatch denoises noisy node matching matrices through $S$ reverse steps. To assess the impact of the number of reverse denoising steps on DiffGED's performance, we evaluate DiffGED using different values of $S$, specifically $S = [20, 10, 5, 4, 3, 2, 1]$. Figure 9 presents the performance comparison across different values of $S$. The results indicate that when $S > 2$, the accuracy and MAE of DiffGED do not vary a lot. However, when $S \leq 2$, accuracy drops significantly while MAE increases. In particular, at $S = 1$, DiffGED becomes a one-shot model, suffering from the same limitations as DiffGED(w/o diffusion), leading to similarly poor performance. Moreover, when $S$ is doubled, the running time of DiffGED almost doubles as well, as the majority of its computational cost comes from denoising the node matching matrix at each reverse step.

**Greedy vs. Exact Node Mapping Extraction.** To evaluate the effectiveness and efficiency of greedy node mapping extraction, we introduce a variant model, DiffGED(Hungarian), which replaces the greedy extraction method with the exact Hungarian algorithm (Kuhn, 1955). As shown in Table 7, DiffGED with greedy node mapping extraction achieves nearly identical accuracy and MAE to DiffGED(Hungarian) across all datasets, while significantly reducing the computational cost of node mapping extraction. This improvement stems from the fact that DiffMatch generates a high-quality sparse node matching matrix, where most elements in each row and column are close to 0, with only a few elements close to 1. This sparsity enables the greedy extraction method to retrieve node mappings comparable to those obtained by the exact Hungarian algorithm while being much faster. To better illustrate this, we show a simple example graph pair in Figure 10, where $\hat{M}$

Table 7: Evaluation on Node Mapping Extraction Strategy

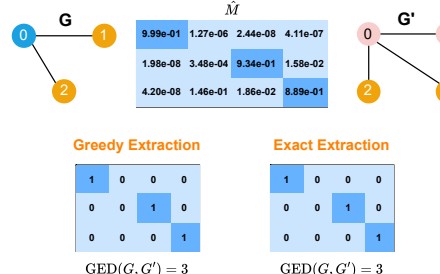

Figure 10: Greedy vs. Exact Node Mapping Extraction

| Datasets | Models | MAE ↓ | Accuracy ↑ | Extraction Time(s) ↓ |
|---|---|---|---|---|
| AIDS700 | DiffGED | 0.022 | 98% | 0.00043 |
| | DiffGED(Hungarian) | 0.021 | 98.1% | 0.0035 |
| Linux | DiffGED | 0.0 | 100% | 0.00036 |
| | DiffGED(Hungarian) | 0.0 | 100% | 0.00345 |
| IMDB | DiffGED | 0.937 | 94.6% | 0.00068 |
| | DiffGED(Hungarian) | 0.918 | 94.7% | 0.00367 |

represents the node matching matrix predicted by DiffMatch. We can see that the predicted $\hat{M}$ is both high-quality and sparse, leading to identical extracted node mappings under both the greedy and Hungarian strategies, resulting in $GED(G, G') = 3$.

## E  THE USE OF LARGE LANGUAGE MODELS (LLMS)

In this paper, LLMs are used solely for polishing the writing.

