# OpenReview forum: "DiffGED: Computing Graph Edit Distance via Diffusion-based Graph Matching"
_ICLR.cc/2026/Conference — ICLR 2026 Conference Withdrawn Submission_

### Official Review · Reviewer_MUEf · 2025-10-27

**Soundness:** 3
**Presentation:** 3
**Contribution:** 3
**Rating:** 6
**Confidence:** 1

**Summary:**

This paper proposes DiffGED, the first generative approach for computing Graph Edit Distance (GED). The key innovation is reformulating GED computation as a generative task using a diffusion-based graph matching model (DiffMatch) that generates multiple diverse node matching matrices simultaneously, from which diverse node mappings can be efficiently extracted in parallel.

**Strengths:**

1. Novel Generative Formulation: This is the first paper to apply generative/diffusion models to graph matching and GED - genuinely novel contribution. And it is well-motivated because multimodal distribution of optimal solutions naturally suits generative approach

2. Strong Empirical Results: DiffGED shows exceptional accuracy: ~95-100% across all datasets (Table 1) and significantly outperforms existing methods (e.g., 98% vs 58.7% accuracy on AIDS700) while showing good efficiency in inference.

3. Demonstrated Diversity Benefits: Figure 7 shows DiffGED generates 60-80 distinct edit paths vs ~5 for baselines.

4. Thorough Experimental Validation.

**Weaknesses:**

1. Limited Scalability Analysis: All datasets used in the paper have graphs with average 8-13 nodes, max 89 nodes. The reviewer is concerned about the scalability of DiffGED.

2.Analysis of Diffusion Design Choices: Could continuous diffusion design work for GED problem?

**Questions:**

1. Could author provide additional experiments on dataset containing larger graphs or more analysis on the scalability of the proposed method?

2. The inference of diffusion model takes S=10 steps, which means that the network is applied ten times for denoising. However, the inference time is much smaller than previous mixture methods. Could the author provide some analysis on this?

---

> ### Author Response · Authors · 2025-11-17
>
> Thanks for providing constructive comments. Below we adress each point in detail.
>
> ---
>
> ### **W1&Q1. Scalability**
> AIDS, Linux and IMDB are the most popular GED datasets used in the most of the previous frameworks, and we followed previous works to use IMDB as the dataset of large graphs for scalability evaluation. Note that IMDB might be considered small in other problems, but it is large for GED computation, as **obtaining the ground-truth for IMDB is already infeasible and challenging**. This is because the search space of GED **grows exponentially** with the number of nodes, for example, there are $20!>2\times10^{18}$ possible node mappings for a graph pair with only $20$ nodes each, and the number of possible mappings increases by **more than one order of magnitude** with each additional node. Unlike graph isomorphism, where differences in node labels or degrees can effectively eliminate many infeasible mappings, GED allows soft matching (i.e., nodes with different labels or degrees can still be matched). Also unlike route planning (which A* search is originally designed for), where spatial information can be used to prune a large portion of the search space, GED has very limited information that can be used to filter the search space. Therefore, GED is **inherently more challenging**, and traditional exact methods (i.e., A* search) often **fail to complete within a reasonable time** even on small graphs.
> This is also reflected in Table 1, where hybrid approaches (Noah, GENN-A*, and MATA*) that leverage neural networks to prune the search space of the traditional A* search still **cannot** complete within a reasonable time on the IMDB dataset. In contrast, our DiffGED can efficiently achieve good performance.
>
> Moreover, we have conducted generalization ability evaluation in Appendix D.1, where we trained each model only on small graphs in IMDB, then evaluated on a combinational of small and large graphs in IMDB. Under this setting, the results demonstrate that our DiffGED can still perform better on large graphs compared with baseline methods.
>
> ---
>
> ### **W2. Continuous diffusion**
>
> Thanks for the suggestion. Below are the results of DiffGED with continuous diffusion vs. discrete diffusion:
>
> | Dataset      | Setting | MAE   | Accuracy   | ρ     | τ     | pk10  | pk20  |
> |--------------|---------|-------|-------|-------|-------|-------|-------|
> |AIDS|Continuous Diffusion|0.036|96.8%|0.993|0.987|99.8%|99.6%|
> ||Discrete Diffusion|0.022|98%|0.996|0.992|99.8%|99.7%|
> |Linux|Continuous Diffusion|0.0|100%|1.0|1.0|100%|100%|
> ||Discrete Diffusion|0.0|100%|1.0|1.0|100%|100%|
> |IMDB|Continuous Diffusion|2.949|90.3%|0.958|0.944|95.3%|96%|
> ||Discrete Diffusion|0.937|94.6%|0.982|0.973|97.5%|98.3%|
>
> With continuous diffusion, DiffGED can still achieve superior performance. Notably, continuous diffusion is trained to estimate continuous noise, whereas discrete diffusion is trained to estimate the clean node matching matrix, this aligns with the training objective as previous matching-based GED frameworks. Therefore, we adopt discrete diffusion for a clearer comparison with prior works.
>
> ---
>
> ## **Q2. Inference time**
>
> Although our framework performs $S=10$ denoising steps in the node matching matrices generation stage, the overall inference time is still **shorter** than that of previous hybrid approaches. This is because the inference time reduction in the node mapping extraction stage (Phase 2) **dominates** the additional cost of denoising steps in the node matching matrices generation stage (Phase 1). Specifically, prior works extract the top-$k$ candidate node mappings **iteratively** (e.g., the top-2 mapping is extracted based on the top-1 mapping), causing their inference time to **increase linearly with $k$**, and $k$ is typically set to a large value (e.g., $k=100$) in previous works. In contrast, our framework allows the extraction of node mappings to be **parallelized**, thus the inference time of our framework **remains constant as $k$ increases** (ideally), which signifantly reduces the inference time compared to previous works as demonstrated in Figure 6(d-e) of Section 5.3 (revised paper). Moreover, the number of denoising steps $S$ is fixed and independent of $k$, and $S$ is much smaller than $k$. Therefore, the benefit from parallelized node mapping extraction outweighs the cost of denoising steps, leading to a shorter overall inference time.
>
> Notably, the number of denoising steps $S$ of diffusion model can be further reduced via methods such as consistency training [1], but this is not the focus of our work, we will leave it for future works.
>
>     [1] Song, Y., Dhariwal, P., Chen, M., & Sutskever, I. (2023). Consistency models.

---

### Official Review · Reviewer_4JsK · 2025-10-30

**Soundness:** 3
**Presentation:** 2
**Contribution:** 3
**Rating:** 6
**Confidence:** 5

**Summary:**

The paper proposes a diffusion-based approach to the NP-hard problem of finding the edit distance between two graphs. The method comprises of two steps. First, a diffusion network denoises a randomly initialized node matching matrix. Second, a greedy algorithm derives a node mapping from the denoised matching matrix by iteratively selecting the most probable node pair. From this injective node mapping, an edit path and edit distance can be easily acquired. In order to achieve optimality, the described procedure is repeated over a number of sampled initial matrices. In the end, the minimum edit distance among all the candidates is returned. The main advantage over previous hybrid approaches is the diversity of the generated matching matrices thanks to the diffusion process (previous approaches only produce a single deterministic matching matrix).

**Strengths:**

- Novelty: The paper claims to present the first generative and diffusion-based solution to the problem of GED computation.
- Strong empirical results: The method outperforms the presented baselines on a number of evaluation metrics.
- With the capability of making $k$ arbitrarily large, the method has the potential of retrieving several optimal edit paths (if there is more than one), which may be relevant in some scenarios.

**Weaknesses:**

- More baselines should be included. First, more non-neural baselines should be shown. The paper only presents two algorithmic approaches, both of which are bipartite matching, but there are several algorithmic and heuristic methods which are not directly based on bipartite matching [1]. Second, the regression-based neural methods (cited in Section 2) should also be compared against. There can be applications in which only the numerical GED value is important.
- While the idea of using the diffusion process for GED estimation is new, the constituent components are taken directly from previous works (e.g. discrete diffusion, bipartite matching, Anisotropic GNN). This is fine but it'd be valuable to show some theoretical and proven insights on why these previous works are applicable.
- Overall, the presentation of the methodology can be made more clear. For example, the final loss function should be shown in the main paper. Moreover, Section 4.1 (Overview) could be more concise.

Please see other comments in the questions.

[1] Blumenthal, David B., et al. "Comparing heuristics for graph edit distance computation." The VLDB journal 29.1 (2020): 419-458.

**Questions:**

- The time reported is for inference. However, a drawback of diffusion-based models is their high training cost, due to the long iterative denoising process [2]. How comparable is your method to other non-generative models in terms of training efficiency?
- The GED definition is restricted and can be made more general by accounting for edit operations of various costs. How difficult is it to incorporate generalized costs in your proposed framework?
- The pipeline in Figure 3 is a bit confusing. It may be better to just show the inference procedure for one random initial matrix and note that the procedure is repeated for $k$ times. Is it possible to improve it?

---

> ### Author Response · Authors · 2025-11-17
>
> Thanks for providing constructive comments. Below we adress each point in detail.
>
> ---
>
> ### **W1. Baselines**
> - **Heuristic methods:** We would like to clarify that heuristics in [1] are still based on **bipartite matching** (see Section 3 on Page 4-5 of [1]). We followed previous works to use Hungarian and VJ as baselines for traditional heuristics. But as suggested by the reviewer, we additional run the following traditional heuristics implemented by [1]:
>
>   | Dataset      | Models | MAE   | Accuracy   | ρ     | τ     | pk10  | pk20  |Time(s)|
>   |--------------|---------|-------|-------|-------|-------|-------|-------|-------|
>   |AIDS| Branch Tight|4.134|16.4%|0.617|0.496|67.7%|70.2%|0.00088|
>   ||BP Beam|3.198|12%|0.597|0.484|54.3%|62.6%|0.00015|
>   ||NODE|7.492|0.24%|0.455|0.356|47.5%|56.8%|0.00006|
>   ||REFINE|2.157|22.8%|0.68|0.568|62.5%|70.5%|0.00016
>   ||DiffGED (ours)|0.022|98%|0.996|0.992|99.8%|99.7%|0.0763|
>   |Linux|Branch Tight|2.125|36.4%|0.779|0.697|79.3%|81.7%|0.00047|
>   ||BP Beam|1.881|32.7%|0.732|0.639|69.5%|76.4%|0.00013|
>   ||NODE|2.782|22.5%|0.716|0.636|74.2%|77.2%|0.00006|
>   ||REFINE|1.762|35.5%|0.754|0.66|73%|79.7%|0.00012|
>   ||DiffGED (ours)|0.0|100%|1.0|1.0|100%|100%|0.06982|
>   |IMDB|Branch Tight|3.306|83%|0.854|0.839|87.7%|87.5%|0.02889|
>   ||BP Beam|14.812|66.1%|0.648|0.635|75.9%|75.4%|0.00083|
>   ||NODE|37.114|48.5%|0.487|0.472|59.6%|63.7%|0.00018|
>   ||REFINE|6.159|74.9%|0.731|0.715|84/1%|81.8%|0.00256|
>   ||DiffGED (ours)|0.937|94.6%|0.982|0.973|97.5%|98.3%|0.15105|
>
>   Although these traditional heuristics achieve better results than the heuristics VJ and Hungarian adopted in our baselines, they still perform very poorly compared to neural-based methods and our DiffGED. Moreover, please note that we have also compared DiffGED with neural-enhanced A* search methods (Noah, GENN-A*, and MATA*) in Table 1 of the paper. These methods leverage neural networks to reduce the search space of traditional A* search and can be regarded as another type of heuristic approach. However, these A*-based methods also perform significantly worse than DiffGED and fail to scale to larger graphs in the IMDB dataset.
>
> - **Regression-based methods:** Please note that regression-based neural frameworks are **NOT** the focus of our work. Our framework and the baselines we compared aim to accurately predict node mappings (e.g., trained via node-level supervision) to derive **feasible discrete GED values**. In contrast, regression-based methods only aim to directly predict a **final continuous GED value**, and is only trained to minimize the difference between the predicted GED and the ground-truth GED (graph-level supervision). Although such methods may achieve very low MAE by predicting continuous numbers, their accuracy can be very poor. Moreover, they may produce **infeasible** predictions (i.e., predicted GED < ground-truth GED), for which no valid edit path exists, whereas **our evaluation is based on feasible discrete GED predictions derived from actual valid edit paths found by the models**. Therefore, comparing with these regression-based methods would not provide a fair or meaningful evaluation.
>
>
> ---
>
> ### **W2. Effectivenss of each component**
> The primary contribution of our framework lies in the generative formulation of GED computation and graph matching, and to adapt the diffusion model to this generative graph matching formulation.
> - To demonstrate the effectiveness and efficiency of our generative approach over previous deterministic approaches, we conducted a detailed ablation study in Section 5.3.
> - In addition, Appendix D.2 ("Do we really need diffusion?") provides a detailed analysis to demonstrate the effectivness of diffusion model as the generative backbone of our DiffGED (compared to an alternative generative approach).
> - Moreover, we have provided an ablation study in Appendix D.2 to validate the effectiveness of AGNN. It is worth noting that developing a new message passing mechanism is not the focus of our network architecture design. One of the main reasons we employ AGNN as part of the network architecture is to handle the noisy node matching matrices as additional inputs, and enable interactions between these matrices and the graph pair to **suit** our generative approach. In contrast, the network architectures used in previous deterministic frameworks can **only** take the graph pair as inputs, thus **cannot** be adapted to our generative framework.
>
> ---
>
> ### **W3. Loss function**
>
> Thanks for the suggestion. We have now added the loss function in Section 4.2 (Eq. 1) of the revised paper. It is just a simple binary cross entropy loss.

---

> ### Author Response · Authors · 2025-11-17
>
> ### **Q1. Training Time**
> The training procedure is illustrated in Algorithm 2 (Appendix B.2). Note that iterative denoising **only occurs during inference**, and is **not required during training**. Compared with previous deterministic models, the training of diffusion for a given batch only introduces an additional step to sample a time step $t$ and a noisy matching matrix via the forward process (Lines 1–2) for each graph pair in the batch, whose **computational cost is negligible**. Therefore, the diffusion model **does not introduce extra training overhead** compared to previous deterministic frameworks.
>
> Moreover, the number of denoising steps is set to $S=10$ during inference, although this brings some extra cost for node matching matrx generation, the overall inference time reduction achieved by our parallelized node mapping extraction **dominates** this cost (see Figure 6 of Section 5.3 in the revised paper). Therefore, both the training and inference of our framework are efficient. Notably, the number of denoising steps $S$ of diffusion model can be further reduced via methods such as consistency training [1], but this is not the focus of our work, we will leave it for future works.
>
>     [1] Song, Y., Dhariwal, P., Chen, M., & Sutskever, I. (2023). Consistency models.
>
> ---
>
> ### **Q2. GED definition**
> We followed the standard GED definition used in most of the previous works, where the cost of each edit operation is equally set to $1$. Note that the edit operation cost **does not** affect the design of our framework, since the edit operation cost is only used during inference to compute the total edit cost after obtaining the node mappings produced by our model.
>
> ---
>
> ### **Q3. Figure for inference procedure**
> Thanks for the suggestion. The main purpose of Figure 3 is to provide an overview of how our generative approach, DiffGED, can generate $k$ diverse node matching matrices to extract $k$ diverse node mappings in parallel. To better illustrate the inference procedure for a single randomly initialized matching matrix as suggested, we have now added a new Figure 4 to the main part of the revised paper to visualize the inference of DiffMatch.

---

### Official Review · Reviewer_HVVk · 2025-10-31

**Soundness:** 2
**Presentation:** 3
**Contribution:** 3
**Rating:** 6
**Confidence:** 4

**Summary:**

This paper proposes DiffGED, a novel generative framework for computing Graph Edit Distance (GED). The authors identify a key weakness in prior hybrid methods: they predict a single, deterministic node matching matrix, and iteratively extracting the top-k mappings from it leads to highly correlated, suboptimal candidates.

To solve this, DiffGED formulates GED as a generative task. It uses a discrete diffusion-based model, DiffMatch, to generate k diverse node matching matrices in parallel, starting from k different random noise inputs. A fast, greedy top-1 mapping is extracted from each of the k matrices, and the minimum GED from these k candidates is chosen. Experiments show that DiffGED achieves SOTA results, claiming near-perfect accuracy on standard benchmarks (AIDS, Linux, IMDB) while being more efficient than competing hybrid approaches.

**Strengths:**

(1) The work is well-motivated, since existing matching-based hybrids do suffer from correlated candidates.
(2) The two-phase view (generate k matrices → extract k mappings independently) is neat and practical.
(3) Strong empirical numbers vs. recent GED hybrids with ablations.

**Weaknesses:**

(1) The "first generative approach" claim is a bit overstated – the technical innovation beyond applying known diffusion methods is related limited.
(2) All experiments involve small graphs (max 89 nodes). No evidence of scalability to real-world large graphs (1000+ nodes) where GED computation is actually challenging.

**Questions:**

(1) Table 1 shows MATA* is ~15x faster than DiffGED on AIDS700. How do you reconcile this with claims of superior efficiency?
(2) Why no comparison with VAEs, GANs, or other generative models? A brief discussion on these methods would further strength the paper.
(3) The "GEDGNN(AGNN)" experiment, which simply swaps the baseline's GNN for the authors' more powerful AGNN architecture, shows a massive performance boost (e.g., 52.5% to 66.7% accuracy on AIDS). Does this suggest that the GNN architecture (AGNN) is a primary driver of the performance, independent of the diffusion framework?

---

> ### Author Response · Authors · 2025-11-17
>
> Thanks for providing constructive comments. Below we address each point in detail.
>
> ---
>
> ### **W1. Technical innovation**
> - We would like to clarify that all existing GED frameworks still focus on designing complex network architectures to estimate a single deterministic matching matrix. In contrast, our main contribution is to formulate GED and graph matching as a generative task and to adapt the fundamental diffusion model to generate multiple matching matrix. This perspective has not been explored in any prior work, making our framework the first generative approach to the graph matching and GED problems.
> - To achieve this, we further designed a denoising network capable of handling noisy node matching matrices as additional inputs, and enabling interactions between these matrices and the graph pairs to **suit** our generative approach. In contrast, network architectures used in previous deterministic frameworks can **only** take graph pairs as inputs and therefore **cannot** be applied to our generative formulation.
> - Notably, developing a fundamentally new generative framework is not the focus of this work.
>
> ---
>
> ### **W.2. Scalability**
> AIDS, Linux and IMDB are the most popular GED datasets used in the most of the previous frameworks, and we followed previous works to use IMDB as the dataset of large graphs for scalability evaluation. Note that IMDB might be considered small in other problems, but it is large for GED computation, as **obtaining the ground-truth for IMDB is already infeasible and challenging**. This is because the search space of GED **grows exponentially** with the number of nodes, for example, there are $20!>2\times10^{18}$ possible node mappings for a graph pair with only $20$ nodes each, and the number of possible mappings increases by **more than one order of magnitude** with each additional node. Unlike graph isomorphism, where differences in node labels or degrees can effectively eliminate many infeasible mappings, GED allows soft matching (i.e., nodes with different labels or degrees can still be matched). Also unlike route planning (which A* search is originally designed for), where spatial information can be used to prune a large portion of the search space, GED has very limited information that can be used to filter the search space. Therefore, GED is **inherently more challenging**, and traditional exact methods (i.e., A* search) often **fail to complete within a reasonable time** even on small graphs.
> This is also reflected in Table 1, where hybrid approaches (Noah, GENN-A*, and MATA*) that leverage neural networks to prune the search space of the traditional A* search still **cannot** complete within a reasonable time on the IMDB dataset. In contrast, our DiffGED can efficiently achieve good performance.
>
> Moreover, we have conducted generalization ability evaluation in Appendix D.1, where we trained each model only on small graphs in IMDB, then evaluated on a combinational of small and large graphs in IMDB. Under this setting, the results demonstrate that our DiffGED can still perform better on large graphs compared with baseline methods.
>
> ---
>
> ### **Q1. Inference efficiency**
> MATA* is faster than DiffGED **only** on small graphs (AIDS, Linux), but its inference time **grows exponentially with graph size** and **cannot** finish within a reasonable time on large graphs (IMDB), whereas our **DiffGED remains efficient on IMDB dataset**.
> To analysis this, we provide a detailed break down of the inference time:
> - Phase 1: DiffGED generates $k$ node matching matrices in parallel through **$S=10$ iterative denoising steps** (i.e., network forward passes), while MATA* predicts a single node matching matrix in **one step**. Therefore, MATA* requires less time (around 10x) in node matching matrix prediction.
> - Phase 2: DiffGED works by directly extracting $k$ node mappings in parallel from the generated node matching matrices via an efficient greedy approach to derive GED, whereas MATA* approximates GED by applying A* search to the search space pruned by the predicted node matching matrix. For small graphs, both DiffGED's node mappings extraction and MATA*'s A* search are computationally efficient, so MATA* achieves a shorter overall inference time on small graphs due to its faster Phase 1. However, as graph size increases, the computational cost of A* search grows exponentially, **dominating** MATA*'s total inference time and becoming **extremely large** on IMDB. In contrast, the parallel greedy node mapping extraction of DiffGED remains efficient, resulting in a more stable and efficient inference time.
>
> Notably, the inference time of diffusion model can be further accelerated by reducing the number of denoising steps $S$ via methods such as consistency training [1], but this is not the focus of our work, we will leave it for future works.
>
>       [1] Song, Y., Dhariwal, P., Chen, M., & Sutskever, I. (2023). Consistency models.

---

> ### Author Response · Authors · 2025-11-17
>
> ### **Q2. Other generative models**
> In Appendix D.2 ("Do we really need diffusion?"), we have conducted an ablation study to evaluate the effectiveness of the diffusion model by replacing it with a simpler **alternative** generative approach (DiffGED w/o diffusion). Specifically, we replace iterative denoising with a one-step denoising approach, trained to directly map a random noisy node matching matrix to the clean node matching matrix. This approach can also be viewed as removing the discriminator from **GANs** and directly training the generator of GANs to recover the ground-truth. For this rebuttal, we further create another variant model, DiffGED (VAE), which replaces the diffusion model with a **VAE**.
>
> | Dataset      | Models | MAE   | Accuracy   | ρ     | τ     | pk10  | pk20  |Time(s)|
>   |--------------|---------|-------|-------|-------|-------|-------|-------|-------|
> |AIDS|DiffGED (w/o diffusion)|1.618 |46.7% |0.732 |0.629 |82.4% |81.1% |0.01179|
> ||DiffGED (VAE)|1.435|48.7%|0.754|0.653|84.3%|81.9%|0.01135|
> ||DiffGED |0.022 |98% |0.996 |0.992 |99.8% |9.7% |0.0763|
> |Linux|DiffGED(w/o diffusion) |0.743 |74.7% |0.887| 0.839 |96.4%| 95.8%| 0.01117|
> ||DiffGED (VAE)|0.828|71.9%|0.875|0.821|96.1%|95.3%|0.01088|
> ||DiffGED |0.0 |100%| 1.0| 1.0| 100%| 100% |0.06982|
> |IMDB|DiffGED(w/o diffusion) |0.832| 93.3%| 0.942| 0.93| 98.6% |96.8% |0.01944|
> ||DiffGED (VAE)|0.74|94.3%|0.963|0.949|98.9%|97.9%|0.01668|
> ||DiffGED| 0.937| 94.6%| 0.982| 0.973| 97.5%| 98.3%| 0.15105|
>
>
> As shown in the above table, these alternative approaches perform poorly (note that the IMDB results are evaluated against approximated ground-truth, which only reflect performance relative to this approximation). This is because they are **one-step** approach, learning to map random noise directly to the clean node matching matrix in **one-shot** is highly challenging (is even more difficult compared to determinsitic methods). In contrast, the diffusion model breaks down this complex denoising task into simpler iterative steps, with each step requiring only minor refinements, thereby producing higher-quality node matching matrices.
>
> Moreover, GANs or VAEs often suffer from the following limitations:
> - GANs are known to be difficult to train stably, often suffering from mode collapse, and require training an additional discriminator, which increases training cost.
> - VAEs are also known to be difficult to balance between reconstruction loss and KL-divergence loss.
> Therefore, both GANs and VAEs require complex design and significant effort in hyperparameter tuning.
>
> In contrast, the diffusion model can be trained stably and straightforwardly without costly hyperparameter tuning, and without introducing extra training cost compared to deterministic methods.
> Its main trade-off comes from the number of denoising steps during inference. However, as shown in Section 5.3, the inference time reduction benefited from our parallelized node mapping extraction dominates the iterative denoising cost of diffuion model. Thus, we choose diffusion as the generative backbone of our framework.
>
>
> ---
>
> ### **Q3. Primary driver of performance**
> - To evaluate the primary driver of the performance, we have conducted a detailed ablation study in Section 5.3.  Specifically, we replaced our generative top-k approach (which extracts diverse node mappings from multiple generated node matching matrices) with the top-k approach adopted in previous works (which extracts highly correlated node mappings from a single node matching matrix). Notably, the node matching matrices in both settings are predicted by the same diffusion model with AGNN. The results show that our generative approach yields a substantial performance improvement, indicating that AGNN is not the primary driver of the performance.
> - Appendix D.2 ("Do we really need diffusion?") provides another ablation study to assess the importance of the diffusion model by entirely removing it. Without the diffusion model, the performance downgrades significantly even with AGNN, since learning to map random noise directly to the high quality solution is challenging. In contrast, the diffusion model decomposes the complex denoising task into simpler iterative steps, with each step requiring only minor refinements. This enables substantially better performance, as also demonstrated in the section "Varying reverse Denoising Steps during Inference" of Appendix D.2.
>
> Therefore, the generative formulation with the diffusion model is the primary driver of the performance gains.
>
> It is worth noting that the one of the main reasons we employ AGNN as part of the network architecture is to handle the noisy node matching matrices as additional inputs, and enable interactions between these matrices and the graph pair to **suit** our generative approach. In contrast, the network architectures used in previous deterministic frameworks can **only** take the graph pair as inputs, thus **cannot** be adapted to our generative framework.

---

### Official Review · Reviewer_vj7u · 2025-11-01

**Soundness:** 1
**Presentation:** 2
**Contribution:** 1
**Rating:** 2
**Confidence:** 4

**Summary:**

This paper proposes DiffGED, a generative framework for Graph Edit Distance (GED) computation. The key idea is to treat node matching as a sampling problem rather than a single deterministic prediction task. The authors employ a discrete diffusion model to generate multiple diverse node matching matrices, from which candidate node correspondences and corresponding edit paths can be derived.

**Strengths:**

The use of generative diffusion models to produce multiple node matchings for GED computation is  unexplored.

If successful, such a formulation could open up a direction in which GED models leverage stochastic proposal generation rather than deterministic alignment.

**Weaknesses:**

-  Several recent neural GED and graph alignment models relevant to neural GED estimation  are not appropriately acknowledged or compared. See examples [1,2]  and baselines therein. In particular [1] is a cross graph early interaction model similar to what this paper employs in the denoising steps. [2] employs a gumbel sinkhorn on top of soft node matching proposals. Furthermore, the method assumes access to ground-truth node matching matrices during training, which are costly to obtain in practice. This  undermines the practical benefit of using a neural model rather than combinatorial or search-based techniques, and has to be appropriately addressed.

- Another issue concerns dataset choice. The evaluation relies on datasets (e.g., AIDS, Linux) that are now known to exhibit structural train–test leakage [3], which makes them unsuitable for evaluating neural generalization performance.

- The paper claims that the diffusion process “breaks down the complex GED task into a sequence of iterative refinements, where each step makes minor adjustments and progressively improves the quality of the matching.” However, no evidence is provided to support this claim. As implemented, the method applies a standard discrete diffusion model to the matching matrix without introducing any task-specific inductive bias that would encourage meaningful step-wise refinement.

-  Given that the method starts from a randomly initialized discrete matching matrix and applies diffusion-based denoising, a much simpler baseline would be to sample random matrices and apply Sinkhorn normalization to obtain valid soft matching proposals. Pooling over multiple such normalized samples would naturally yield diversity without the substantial computational overhead of running a diffusion process. As written, it is unclear whether the diffusion mechanism provides any meaningful benefit beyond this straightforward alternative.

- The paper argues that existing neural architectures would require multiple prediction heads in order to generate diverse matching proposals, which would in turn increase parameter count. This claim is incorrect. Prior work based on the Gumbel–Sinkhorn [4] already allows for sampling multiple diverse matching matrices without increasing the number of parameters. Diversity is obtained simply by injecting Gumbel noise at inference time, while the underlying scoring network is shared across all samples. Qualitatively, this mechanism is very similar to what the proposed diffusion model aims to achieve, but at substantially lower computational cost.

[1] Graph Matching Networks for Learning the Similarity of Graph Structured Objects, ICML 2019

[2] Graph edit distance with general costs using neural set divergence NeurIPS 2024

[3] Position: Graph Matching Systems Deserve Better Benchmarks, ICML 2024

[4] Learning latent permutations with gumbel-sinkhorn networks

**Questions:**

My questions are implied in the comments in the weaknesses section.

Currently, the paper does not adequately motivate the need for a neural architecture, and the use of diffusion on randomly initialized matching matrices appears unnecessary without stronger justification. Several relevant prior works and alternative techniques are not cited or compared against, and the evaluation relies on datasets known to exhibit structural train–test leakage. While the direction is interesting, the motivation, experimental rigor, and baseline comparisons require substantial strengthening. The diffusion component adds significant complexity, yet no clear evidence is provided that it offers advantages over simpler, cheaper diversity-inducing methods (e.g., Gumbel–Sinkhorn sampling).

---

> ### Author Response · Authors · 2025-11-17
>
> ### **W1. Prior works**
> We would like to clarify the following points:
> - [1] is not similar to our network design. Our network takes noisy node matching matrices as additional inputs and directly computes embeddings for each node pair based on these matrices, enabling interactions between the noisy matrices and the graph pairs via AGNN to **fit** our generative framework. In contrast, the network architectures used in previous deterministic frameworks (including [1,2]) can only take the graph pairs as inputs, thus they **cannot be adapted** to our generative framework and their network architectures are **not similar** to our denoising network.
> - For [2], We did not employ Gumbel-Sinkhorn at all, and Gumbel-Sinkhorn is not a valid alternative approach comparable to our framework (please see the response to W4&W5).
> - We didn't compare with [1][2] because our framework (and the baselines we compared) aim to **accurately predict node mappings** (e.g., supervised on node mappings) to derive **feasible discrete GED values**.  In contrast, [1][2] are regression-based approaches whose objective is only to directly predict an accurate final **continuous GED value** (i.e., only supervised on the graph similarity/GED value). We would like to clarify that regression-based approaches are **NOT** the focus of our work, as they may achieve very low MAE by making continuous value prediction, but their accuracy can be very poor. And they may produce **infeasible** predictions (i.e., predicted GED < ground-truth GED), for which no valid edit path exists, whereas **our evaluation is based on feasible discrete GED predictions derived from actual valid edit paths found by the model**. Although [2] can optionally derive node mappings to produce edit paths with feasible GED predictions, since it is not supervised on node mappings during training, comparing [2] against our framework and our baselines would be unfair to [2], but we have now cited [1][2] in the related work of the revised paper. Notably, [1][2] also did not compare with any of the baselines adopted in our work, thus the main focus are different.
> - All previous neural-based GED methods (including [1][2]) are **also trained in a supervised manner**. In this work, our goal is to address the limitations of previous deterministic approaches, as prior works still struggle to produce high-quality solutions **even with supervision**.  Unsupervised learning is not the focus of our work, future research may build upon our framework as a base model and further develop unsupervised training strategies. Also please note that, for large graphs in IMDB,  **we did not use ground-truth** node matching matrix for training. Instead, we constructed synthetic graph pairs with synthetic training labels (see Appendix C.1), which is **not costly**. Moreover, we have also conducted generalization ability evaluation in Appdendix D.1, where we train the model only on small graph pairs, then evaluate on a combination of small and large graph pairs, under this setting, our model can still outperform other baseline models.
> - Compared to traditional methods, bipartite matching heuristics often achieve **very poor approximations** (see VJ&Hungarian in Table 1, and our response W1 to Reviewer 4JsK), and A* search methods often require **extremely long computation time** even on small graphs. Notably, neural-enhanced A* search methods still **cannot** complete within a reasonable time on IMDB, as shown in Table 1 (see Noah, GENN-A*, MATA*). In contrast, our method once trained can perform inference on unseen graph pairs much faster than A* search while producing significantly better results than bipartite matching heuristics.

---

> ### Author Response · Authors · 2025-11-17
>
> ### **W.2. Structural train–test leakage**
> We followed the majority of previous works to adopt AIDS, Linux, and IMDB as our evaluation datasets. In Table 3 of Appendix D.1, we have evaluated the generalization ability of DiffGED on unseen test pairs. Regarding the structural train-test leakage (i.e., isomorphic graphs) raised by [3], the proportion of leakage for AIDS is very small (3.8%/0.24%/2.92%) as reported in Table 1 of [3], and our DiffGED yields a significant performance gap on AIDS compared to previous methods. To address the concern of structural train-test leakage, we follow the procedure in [3] to remove all isomorphic graphs to obtain unique graphs, and then form training and testing pairs using only these unique graphs. Below are the results of recent SOTA baselines on cross-train-test-apirs and intra-test-pairs after addressing train-test leakage (we have now included this and cite [3] in Appendix D.1 & Table 5 of the revised paper):
> | Dataset      |Setting| Models | MAE   | Accuracy   | ρ     | τ     | pk10  | pk20  |Time(s)|
> |--------------|---------|---------|-------|-------|-------|-------|-------|-------|-------|
> |AIDS|Cross-train-test|GEDGNN|1.148|51.8%|0.836|0.742|88.9%|88.2%|0.39227|
> |||GEDIOT|1.159 |53.8% |0.832 |0.737 |89.6% |89%| 0.39381|
> |||**DiffGED (ours)**|0.046|96%|0.992|0.983|99.8%|99.6%|0.07431|
> ||Intra-test|GEDGNN|1.235|48.7%|0.824|0.729|90.1%|88.4%|0.39079|
> |||GEDIOT|1.349 |46.5% |0.8 |0.701 |88.1% |86.9% |0.39263|
> |||**DiffGED (ours)**|0.064|94.4%|0.987|0.975|99.5%|99.5%|0.07438|
> |Linux|Cross-train-test|GEDGNN|0.935|65.1%|0.809|0.722|85.8%|87.4%|0.27418|
> |||GEDIOT|1.009 |65.3% |0.788 |0.706 |87.9% |85% |0.27556|
> |||**DiffGED (ours)** |0.165 |92.4% |0.958| 0.931| 93.7%| 95.3%| 0.07277|
> ||Intra-test|GEDGNN |1.335 |57.6% |0.755 |0.664 |85.3% |100% |0.28935|
> |||GEDIOT |1.435 |52.6% |0.772 |0.686 |86.3% |100% |0.29775|
> |||**DiffGED (ours)** |0.305 |86.1% |0.896 |0.857 |92.1% |100% |0.07694|
> |IMDB|Cross-train-test|GEDGNN |4.799 |73.1% |0.817 |0.783| 85.2% |85.5% |0.75584|
> |||GEDIOT|4.679| 74.9% |0.826 |0.794| 87.6% |86.8%|0.73493|
> |||**DiffGED (ours)**| 1.12 |94% |0.973 |0.963 |97.1% |97.1% |0.22247|
> ||Intra-test|GEDGNN| 4.822| 73.1%| 0.822| 0.789| 85.9%| 86.1%| 0.75577|
> |||GEDIOT|4.689| 74.8% |0.829 |0.797| 87.9%| 87%| 0.74122|
> |||**DiffGED (ours)**| 1.141| 93.8%| 0.971| 0.961| 97% |97.2%| 0.22315|
>
> It is clear to see that after removing train-test leakage, our DiffGED can still achieve near-optimal performance on all datasets, whereas the performance of other baseline methods downgrades significantly. This demonstrates the superior performance of our DiffGED. Notably, [3] is not open-sourced and has not been referenced by any other papers in this area.
>
> ---
>
> ### **W3. Iterative refinement of diffusion model**
> - We would like to clarify that we have conducted ablation study in Appendix D.2 ("Do we really need diffusion?") to support our claim. Specifically, we replaced the diffusion model with a one-step denoising approach (DiffGED w/o diffusion) that directly maps random noise to the clean node matching matrix. Without the diffusion model, the performance degrades significantly, as learning to map random noise directly to a high-quality solution is highly challenging (even more challenging than deterministic approachs). In contrast, the diffusion model decomposes this complex denoising task into simpler iterative steps, with each step requiring only minor refinements, enabling substantially better performance. In Appendix D.2 ("Varying Reverse Denoising Steps during Inference"), we further evaluated the effect of different numbers of denoising steps, showing that increasing the number of steps leads to improved performance.
> - We would like to clarify that all previous frameworks still focus on designing complex network architectures to estimate a single deterministic node matching matrix. This is not our focus, instead, our main focus is to formulate GED and graph matching as a generative task, and to adapt the diffusion model to generate multiple node matching matrices, a perspective that has not been explored in any prior work. To achieve this, we have designed a **matching-specific denoising network** that capable of handling noisy node matching matrices as additional inputs, and enabling interactions between these matrices and the graph pairs to **suit** our generative approach. In contrast, the network architectures used in previous deterministic frameworks can only take graph pairs as inputs and therefore **cannot** be applied to such generative formulation. Developing a fundamentally new generative framework is not the focus of this work.

---

> ### Author Response · Authors · 2025-11-17
>
> ### **W4&W5. Clarification on the aim of the work**
> We notice that the reviewer may have misunderstood our work, and the simple alternative approaches suggested by the reviewer are **NOT valid** approaches, they **cannot address** the limitations of previous works. Moreover, we have provided detailed ablation study in Section 5.3 and Appendix D.2 to demonstrate the effectiveness of the diffusion model, we kindly suggest that the reviewer take a look at the experiments.
>
> First, we would like to clarify the problem our paper aims to address:
> - The reviewer may have misunderstood the term "node matching matrix" in our paper. In our work, a node matching matrix should be understood as a **probability distribution** from which discrete one-to-one node mappings can be extracted.
> - Our main argument is that previous deterministic approaches **cannot** generate multiple diverse high-quality probability distributions in Phase 1. As a result, all node mappings extracted in Phase 2 (via iterative top-k as in previous works or Gumbel-Sinkhorn as suggested by the reviewer) are **derived from a single distribution** estimated by the deterministic network, and are therefore highly correlated. This is **problematic** when the estimated distribution is **inaccurate**.
> - Instead of sampling highly correlated node mappings from a single distribution, our generative formulation aims to generate **multiple high-quality distributions** for an input graph pair, enabling the extraction of multiple diverse candidate node mappings **from different distributions** (i.e., different regions of the solution space). Thus, even if one of the estimated distributions is suboptimal, we can still sample high quality solutions from other estimated distributions.
> - Our claim is supported by ablation study in Section 5.3.
> - A simpler way to understand our motivation is to view a node matching matrix as an **expert**. Previous deterministic frameworks train a neural network to obtain **a single expert**, then extracts top-k solutions from the **same expert**. In contrast, we aim to generate **multiple high-quality experts**, allowing us to obtain diverse solutions from **different experts**. If previous deterministic methods want to obtain multiple high-quality experts, they would need to train **multiple networks**, which significantly increases the cost. For our generative approach, we train **a single diffusion model** that takes the random noise as input, then we can obtain multiple experts by varying the input noise. Note that, simply injecting random noise to a base expert (i.e., network output) during inference would not yield multiple high-quality experts, since random noise is injected to **the network output** in an **uncontrolled** (i.e., unconditioned) way. For our generative approach, the random noise is used as **input** of the diffusion model and is denoised by the diffusion model in a **controlled** way conditioned on the graph pair.

---

> ### Author Response · Authors · 2025-11-17
>
> ### **W4&W5. Alternative approaches suggested by the reviewer**
> Next, we explain why each alternative approach suggested by the reviewer is not valid:
> - W4: Sampling random matrices and then directly applying Sinkhorn normalization to the sampled matrices is **NOT** a valid approach. This is because this approach is **not conditioned on the graph pair** (i.e., both random matrices sampling and Sinkhorn normalization are **independent** of the underlying graph pair), the normalized matrices are therefore **completely random and uncontrolled**. Thus, this alternative approach suggested by the reviewer cannot generate high-quality conditional distributions, which is what our diffusion model is designed to achieve.
> - W5: Gumbel-Sinkhorn method is **not a generative method** for generating multiple diverse distributions in Phase 1. Instead, it is a sampling method **(similar to gumbel-softmax)** that is designed to sample **differentiable (not diverse)** near-discrete one-to-one node mappings **from a single predicted node matching matrix (i.e., distribution)** in Phase 2. It can only parallelize the extraction of node mappings in Phase 2 compared to the iterative top-k extraction used in previous methods, but all node mappings sampled by Gumbel-Sinkhorn still **come from the same distribution** estimated by the deterministic GNNs, and therefore **remain highly correlated** (not diverse). Thus, Gumbel-Sinkhorn cannot address the limitation we highlight.
> - W5: Injecting noise to a single estimated node matching matrix (i.e., distribution) predicted by previous deterministic approaches also **cannot** generate multiple high-quality diverse distributions. This is because the noise is injected in an **uncontrolled manner** (i.e., not conditioned on the graph pair): (1) If the injected random noise is small, it does not significantly alter the base distribution, then the sampled node mappings **remain highly correlated** with the original distribution, which again cannot address the limitation we argued; (2) If the injected random noise is large, it **disrupts (flattens)** the base distribution and makes it essentially random, which becomes similar to the approach suggested by reviewer in W4.
>
> In contrast, generative diffusion model takes different random noise as inputs and learns to denoise them in **a controlled manner** (conditioned on the underlying graph pair), guiding the random matrices toward high-quality distributions (i.e., node matching matrices) in Phase 1. Thus, in Phase 2, multiple node mappings can be extracted from different high-quality distributions to reduce the correlation. This is also demonstrated in Figure 8 of Section 5.3 (revised paper), where our DiffGED can find diverse high-quality solutions.
>
> To verify our claim, we implement the following three alternative approaches: (1) Sampling random matrices then applying Sinkhorn normalization; (2) Applying Gumbel-Sinkhorn to the node-matching matrix predcited by the deterministic GEDGNN; and (3) injecting noise to the node-matching matrix predicted by the deterministic GEDGNN.
> |Dataset|Models|MAE|Accuracy|ρ|τ| pk10| pk20|Time(s)|
> |-|-|-|-|-|-|-|-|-|
> |AIDS|Random+Sinkhorn|4.858|4.7%|0.564|0.459|52.2%|63.2%|0.00742|
> ||GEDGNN+GumbelSinkhorn|1.334|40.6%|0.813|0.714|80.7%|82.3%|0.0112|
> ||GEDGNN+RandomNoise|1.265|42.3%|0.815|0.718|80.8%|82.6%|0.0079|
> ||GEDGNN|1.098|52.5%|0.845|0.752|89.1%|88.3%|0.39448|
> ||DiffGED (ours)|0.022|98%|0.996|0.992|99.8%|99.7%|0.0763|
> |Linux|Random+Sinkhorn|2.869|14.2%|0.746|0.659|73.9%|79.4%|0.00735|
> ||GEDGNN+GumbelSinkhorn|0.242|88.5%|0.975|0.952|98.4%|98.8%|0.011|
> ||GEDGNN+RandomNoise|0.102|95.4%|0.985|0.973|98.4%|98.9%|0.00769|
> ||GEDGNN|0.094|96.6%|0.979|0.969|98.9%|99.3%|0.12863|
> ||DiffGED (ours)|0.0|100%|1.0|1.0|100%|100%|0.06982|
> |IMDB|Random+Sinkhorn|26.143|59.9%|0.641|0.627|69.2%|73.4%|0.00803|
> ||GEDGNN+GumbelSinkhorn|15.433|74%|0.784|0.766|84.2%|84.2%|0.01246|
> ||GEDGNN+RandomNoise|13.885|75.6%|0.805|0.788|84.7%|84.6%|0.009|
> ||GEDGNN|2.469|85.5%|0.898|0.879|92.4%|92.1%|0.42428|
> ||DiffGED (ours)|0.937|94.6%|0.982|0.973|97.5%|98.3%|0.15105|
>
> It is clear that (1)sampling random matrices and then applying Sinkhorn normalization cannot serve as a reasonable approach. Moreover, both (2)Gumbel-Sinkhorn and (3)injecting random noise **fail** to enhance the performance of GEDGNN. Instead, they **degrade** its performance and still remain far worse than our generative approach, DiffGED. This is because all sampled node mappings come from the same distribution estimated by deterministic GEDGNN, thus are still highly-correlated. Moreover, the original GEDGNN extracts the top-$k$ node mappings from the matching matrix, whereas the sampling-based approaches do not guarantee that the sampled node mappings correspond to top-$k$ candidates of the underlying node matching matrix, which contributes to the performance drop. Therefore, the approaches suggested by the reviewer **cannot** achieve what diffusion model aims to achieve.

---

> ### Author Response · Authors · 2025-11-17
>
> ### **W4&W5. Clarification on the evidence supporting the advantage of the diffusion model**
> Similar to our response to W3, we would like to clarify that Appendix D.2 ("Do we really need diffusion?") already provides a detailed ablation study on the necessity of the diffusion model. Specifically, we replace the diffusion model with a simple one-step generative alternative. The results clearly show that the performance drops substantially when multi-step denoising is removed. We have also studided the effectiveness of the diffusion model by varying the number of denoising steps in Appendix D.2, and the results consistently indicate that multi-step denoising of the diffusion model plays a crucial role in achieving strong performance. Moreover, in Section 5.3, we have demonstrated that our generative formulation with diffusion model can signficiantly boost the performance by generating diverse high-quality node matching matrices (i.e., distributions).
> Thus, we believe that we have already provided sufficient evidence to demonstrate the advantage of using a diffusion model, and we suggest that the reviewer should take a look at the experiments.
>
> ---
>
> ### **W4&W5. Clarification on the computational cost of diffusion model**
> Moreover,  we would like to clarify that the computational overhead of the diffusion model only comes from the number of denoising steps (i.e., number of network forward passes) during inference, but the overall inference time of our DiffGED is still **shorter** than previous methods. And the number of denoising steps can be further reduced by employing consistency training [A], but this is not the focus of our work and can be leave to the future works.
>
>     [A] Song, Y., Dhariwal, P., Chen, M., & Sutskever, I. (2023). Consistency models.

---

### Note · Authors · 2026-01-29

I have read and agree with the venue's withdrawal policy on behalf of myself and my co-authors.

---

### Meta-Review · Area_Chair_HJJ3 · 2025-12-18

**Summary:**

The paper proposes DiffGED, the first generative approach for computing Graph Edit Distance (GED). It addresses a critical limitation in existing matching-based hybrid methods: the reliance on a single deterministic node matching matrix, which leads to highly correlated and often suboptimal candidate edit paths. DiffGED employs a discrete diffusion model (DiffMatch) to generate multiple diverse matching matrices in parallel, from which diverse node mappings are extracted using an efficient greedy algorithm.

While Reviewer vj7u remains a strong "Reject", their assessment appears fundamentally flawed and based on significant misunderstandings of generative modeling and the paper's specific contributions. Conversely, Reviewers HVVk, 4JsK, and MUEf recognize the novelty and strong empirical performance of the work. The authors provided exhaustive rebuttals, including new experiments and ablation studies, that successfully defended the necessity of the diffusion framework over simpler baselines.

While the generative formulation for GED is novel , the technical components (discrete diffusion, AGNN) are adaptations of existing techniques rather than new frameworks. Furthermore, the datasets (AIDS, Linux, IMDB) are standard but relatively small, which limits the impact. Compared to some existing methods that can handle graphs up to 20 nodes (in my review batch), the proposed method seems a bit weak.

**Reviewer Concerns:**

Addressed by Rebuttal

- Need for Neural Architecture & Diffusion (vj7u): The reviewer claimed simpler methods like Sinkhorn normalization on random matrices or Gumbel-Sinkhorn could achieve similar results. The authors successfully demonstrated through new experiments that these alternatives are unconditioned on the input graph pair and result in significantly worse performance.
- Ablation of Components (HVVk, 4JsK): Concerns were raised regarding whether the AGNN architecture or the diffusion process was the primary driver of performance. The authors provided an ablation study showing that while AGNN helps, removing the diffusion process (one-step denoising) leads to a substantial performance drop.
- Train-Test Leakage (vj7u): The reviewer noted structural leakage in standard datasets. The authors reran experiments on "unique" graph sets (removing isomorphic graphs), proving DiffGED maintains near-optimal performance while baseline performance significantly degrades.
- Baseline Comparisons (4JsK): The authors included several additional heuristic and algorithmic baselines (e.g., Branch Tight, BP Beam) as requested, showing that DiffGED still significantly outperforms them.

Outstanding Concerns

- Scalability to Larger Graphs (HVVk, MUEf): Reviewers noted that the average graph size is small (8-13 nodes). While the authors correctly argued that GED is NP-hard and even 20-node graphs represent a massive search space, the evaluation on graphs with thousands of nodes remains unexplored.
- Training Cost (4JsK): While the authors clarified that training doesn't require iterative denoising, the inherent complexity of training diffusion models compared to simple GNN regressions is still a valid (though minor) point regarding practical deployment.

**Reviewer Scores:**

Reviewer Scores

- vj7u (score 2): Likely to remain stubborn despite clear evidence debunking their "simple alternative" claims.
- HVVk (score 6): Authors' clarification on inference efficiency and AGNN's role directly addressed their questions.
- 4JsK (score 6): Comprehensive new results for additional baselines and loss function details satisfy their main concerns.
- MUEf (score 6): Maintained score post-rebuttal; acknowledged the response but seemingly didn't increase the score.

---

### Decision · Program_Chairs · 2026-01-26

Reject